METHODS AND RESOURCES

# Systematic exploration of *Escherichia coli* phage–host interactions with the BASEL phage collection

Enea Maffei [1]☯, Aisylu Shaidullina [1]☯, Marco Burkolter [1], Yannik Heyer[1], Fabienne Estermann [1], Valentin Druelle [1], Patrick Sauer [1], Luc Willi[1], Sarah Michaelis[2], Hubert Hilbi [2], David S. Thaler[1,3], Alexander Harms [1]*

**1** Biozentrum, University of Basel, Basel, Switzerland, **2** Institute of Medical Microbiology, University of Zürich, Zürich, Switzerland, **3** Program for the Human Environment, Rockefeller University, New York City, New York, United States of America

☯ These authors contributed equally to this work.
* alexander.harms@unibas.ch

**Data Availability Statement:** All data generated or analyzed during this study are included in this published article. The raw data of every replicate experiment underlying the plots shown in Figs 3E,

## Abstract

Bacteriophages, the viruses infecting bacteria, hold great potential for the treatment of multi-drug-resistant bacterial infections and other applications due to their unparalleled diversity and recent breakthroughs in their genetic engineering. However, fundamental knowledge of the molecular mechanisms underlying phage–host interactions is mostly confined to a few traditional model systems and did not keep pace with the recent massive expansion of the field. The true potential of molecular biology encoded by these viruses has therefore remained largely untapped, and phages for therapy or other applications are often still selected empirically. We therefore sought to promote a systematic exploration of phage–host interactions by composing a well-assorted library of 68 newly isolated phages infecting the model organism *Escherichia coli* that we share with the community as the BASEL (BActeriophage SElection for your Laboratory) collection. This collection is largely representative of natural *E. coli* phage diversity and was intensively characterized phenotypically and genomically alongside 10 well-studied traditional model phages. We experimentally determined essential host receptors of all phages, quantified their sensitivity to 11 defense systems across different layers of bacterial immunity, and matched these results to the phages' host range across a panel of pathogenic enterobacterial strains. Clear patterns in the distribution of phage phenotypes and genomic features highlighted systematic differences in the potency of different immunity systems and suggested the molecular basis of receptor specificity in several phage groups. Our results also indicate strong trade-offs between fitness traits like broad host recognition and resistance to bacterial immunity that might drive the divergent adaptation of different phage groups to specific ecological niches. We envision that the BASEL collection will inspire future work exploring the biology of bacteriophages and their hosts by facilitating the discovery of underlying molecular mechanisms as the basis for an effective translation into biotechnology or therapeutic applications.

5F, 6E, 7E, 8D, 9F, 10D, 10G, and 12B–12H as well as all calculations underlying the summary data in these figures are compiled in S1 Data. Genome sequences of BASEL collection phages are available at NCBI GenBank (accessions MZ501046-MZ501113).

**Funding:** This work was supported by Swiss National Science Foundation (SNSF) Ambizione Fellowship PZ00P3_180085 (to AH), SNSF National Center of Competence in Research (NCCR) AntiResist, and a generous contribution of the Stiftung Emilia-Guggenheim-Schnurr of the Naturforschende Gesellschaft in Basel (NGiB; to AH). AS and VD were supported by Biozentrum PhD Fellowships. S.M. and H.H. were supported by the Institute of Medical Microbiology, University of Zürich. The funders had no role in study design, data collection and analysis, decision to publish, or preparation of the manuscript.

**Competing interests:** The authors have declared that no competing interests exist.

**Abbreviations:** Abi, abortive infection; BASEL, BActeriophage SElection for your Laboratory; EAEC, enteroaggregative *E. coli*; ECA, enterobacterial common antigen; EOP, efficiency of plating; ICTV, International Committee on the Taxonomy of Viruses; LB, Lysogeny Broth; LPS, lipopolysaccharide; PBS, phosphate-buffered saline; RBP, receptor-binding protein; RM, restriction–modification; UPEC, uropathogenic *E. coli*.

## Introduction

Bacteriophages, the viruses infecting bacteria, are the most abundant biological entities on earth with key positions in all ecosystems and carry large part of our planet's genetic diversity in their genomes [1–3]. Out of this diversity, a few phages infecting *Escherichia coli* became classical models of molecular biology with roles in many fundamental discoveries and are still major workhorses of research today [4]. The most prominent of these are the seven "T phages" T1 to T7 ([5]; reviewed in [6]) and bacteriophage lambda [7]. Like most known phages, these classical models are tailed phages or Caudovirales that use characteristic tail structures to bind host surface receptors and to eject their genomes from the virion head into the host cell. Three major virion morphotypes of Caudovirales are known, myoviruses with a contractile tail, siphoviruses with a long and flexible tail, and podoviruses with a very short, stubby tail [2]. While the T phages are all so-called virulent phages that kill their host to replicate at each infection event, lambda is a temperate phage and can either kill the host to directly replicate or decide to integrate into the host's genome as a prophage for transient passive replication by vertical transmission in the so-called lysogen [2,8]. These 2 alternative lifestyles as obligately lytic or as temperate phages have major implications for viral ecology and evolution: While virulent phages have primarily been selected to overcome host defenses and maximize virus replication, temperate phages characteristically encode genes that increase the lysogens' fitness, e.g., by providing additional bacterial immunity systems to fight other phages [8,9].

The ubiquity of phage predation has driven the evolution of a vast arsenal of bacterial immunity systems targeting any step of phage infection [9,10]. Inside host cells, phages encounter 2 lines of defense of which the first primarily comprises restriction–modification (RM) systems or CRISPR-Cas, the bacterial adaptive immunity, that directly attack viral genomes [11]. A second line of defense is formed by diverse abortive infection (Abi) systems that protect the host population by triggering an altruistic suicide of infected cells when sensing viral infections [9–12]. While RM systems and CRISPR-Cas are highly abundant and have been successfully adapted for biotechnology (honored with Nobel Prizes in 1978 and 2020, respectively), the molecular mechanisms underlying the function of collectively abundant, but each individually rare, Abi systems have remained elusive with few exceptions [9–12].

Research on phages has expanded at breathless pace over the last decade with a focus on biotechnology and on clinical applications against bacterial infections ("phage therapy") [13,14]. Besides or instead of the few traditional model phages, many researchers now employ comparably poorly described, newly isolated phages that are often only used for a few studies and available only in their laboratory. The consequence of this development is a rapidly growing amount of very patchy data such as, e.g., the currently more than 15,000 available unique phage genomes for which largely no linked phenotypic data are available [15]. Despite the value of proof-of-principle studies and of a rich genome database, this lack of systematic, inter-linked data in combination with the diversity of bacteriophages makes it very difficult to gain a mechanistic understanding of phage biology or to uncover patterns in the data that would support the discovery of broad biological principles beyond individual models.

As an example, phage isolates for treating bacterial infections are necessarily chosen largely empirically due to the lack of systematic data about relevant phage properties. Currently, the selection of native phages and their engineering for therapeutic applications primarily focus on a virulent lifestyle, a broad host range, and very occasionally on biofilm- or cell wall–degrading enzymes that are comparably well understood genetically and mechanistically [13,14,16,17]. However, the molecular mechanisms and genetic basis underlying other desired features such as resistance to bacterial immunity systems and, in general, the distribution of all these features across different groups of phages have remained understudied. Given the notable incidence of

treatment failure in phage therapy [18–20], a better understanding of the links between phage taxonomy, genome sequence, and phenotypic properties seems timely to select more effective native phages for therapeutic applications and to expand the potential of phage engineering.

In this work, we therefore present the BASEL (Bacteriophage Selection for your Laboratory) collection as a reference set of 68 newly isolated virulent phages that infect the laboratory strain *E. coli* K-12 and make it accessible to the scientific community. We provide a systematic phenotypic and genomic characterization of these phages alongside 10 classical model phages regarding host receptors, sensitivity and resistance to bacterial immunity, and host range across diverse enterobacteria. Our results highlight clear phenotypic patterns between and within taxonomic groups of phages that suggest strong trade-offs between important bacteriophage traits. These findings greatly expand our understanding of bacteriophage ecology, evolution, and their interplay with bacterial immunity systems. We therefore anticipate that our work will not only establish the BASEL collection as a reference point for future studies exploring fundamental bacteriophage biology but also promote a rational application of phage therapy based on an improved selection and engineering of bacteriophages.

## Results

### Composition of the BASEL collection

The first aim of our study was to generate a collection of new phage isolates infecting the ubiquitously used *E. coli* K-12 laboratory strain that would provide representative insight into the diversity of virulent tailed phages by covering all major groups and containing a suitable selection of minor ones. We therefore started by generating a derivative of *E. coli* K-12 without any of the native barriers that might limit or bias phage isolation unfavorably. This strain, *E. coli* K-12 MG1655 ΔRM, therefore lacks the O-antigen glycan barrier (see below), all restriction systems, as well as the RexAB and PifA Abi systems (see Materials and methods as well as S1 Text). *E. coli* K-12 ΔRM was subsequently used to isolate hundreds of phages from environmental samples such as river water or compost, but mostly from the inflow of different sewage treatment facilities (Fig 1A and Materials and methods).

Previous bacteriophage isolation studies had already provided deep insight into the diversity of tailed, lytic *E. coli* phages in samples ranging from sewage over diverse natural environments to infant guts or blood and urine of patients in tertiary care [21–27]. Despite the wide diversity of known *E. coli* phages [28], nearly all phage isolates reported in these studies belonged to 5 major groups: Among myoviruses, the so-called "T-even" phages (relatives of T2, T4, and T6; Myoviridae subfamily Tevenvirinae) dominated together with relatives of *E. coli* O157:H7 typing phage rV5 (Myoviridae subfamily Vequintavirinae and related phages). Frequently isolated groups of siphoviruses were relatives of phage T5 (Demerecviridae subfamily Markadamsvirinae), the highly diverse kin of phage T1 (Drexlerviridae family), and the genera *Dhillonvirus*, *Seuratvirus*, and *Nonagvirus* of Siphoviridae. Podoviruses of any kind were rarely reported, and if, then were mostly relatives of T3 and T7 (Autographiviridae subfamily Studiervirinae) isolated using enrichment cultures that are known to greatly favor such fast-growing phages [22,29]. This pattern does not seem to be strongly biased by any given strain of *E. coli* as isolation host because a large, very thorough study using diverse *E. coli* strains reported essentially the same composition of taxonomic groups [21].

Whole-genome sequencing of around 120 different phage isolates from our isolation experiments largely reproduced this pattern, which suggests that several intrinsic limitations of our approach did not strongly affect the spectrum of phages that we sampled (S2 Text). The virion morphology of representative isolates of each taxonomic group was studied by transmission electron microscopy to confirm the sequence-based classification. After eliminating closely

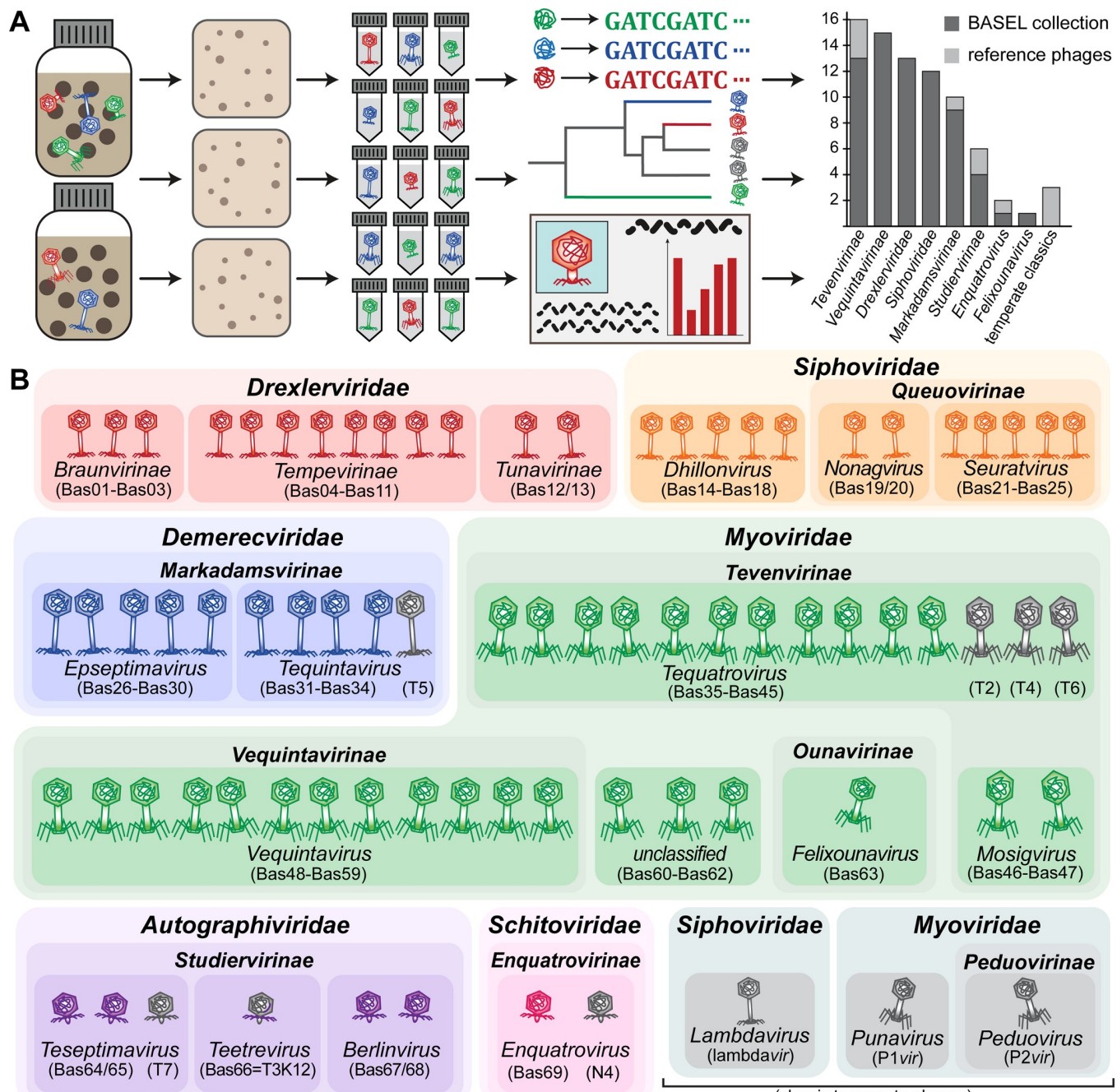

**Fig 1. Overview of the BASEL collection. (A)** Illustration of the workflow of bacteriophage isolation, characterization, and selection that resulted in the BASEL collection (details in Materials and methods; the bar diagram includes phi92-like phages in Vequintavirinae for simplicity). **(B)** Taxonomic overview of the bacteriophages included in the BASEL collection and their unique Bas## identifiers. Newly isolated phages are colored by current taxonomic classification, while reference phages are shown in gray. BASEL, BActeriophage SElection for your Laboratory.

related isolates, the BASEL collection was formed as a set of 68 new phage isolates (Fig 1B; see also Materials and methods and S5 Table). We deliberately did not give full proportional weight to highly abundant groups like T-even phages so that the BASEL collection is not truly representative in a narrow quantitative sense. Instead, we included as many representatives of rarely isolated groups as possible to increase the biological diversity that could be an important asset for studying the genetics of phage/host interactions or for unraveling genotype/phenotype relationships. Besides these 68 new isolates (serially numbered as Bas01, Bas02, etc.; Fig 1 and S5 Table), we included a panel of 10 classical model phages in our genomic and phenotypic characterization and view them as an accessory part of the BASEL collection. Beyond the T phages (except T1 that is a notorious laboratory contaminant [30]), we included well-studied podovirus N4 and obligately lytic mutants of the 3 most commonly studied temperate phages lambda, P1, and P2 [5,7,31,32] (Fig 1; see also S5 Table).

## Overview: Identification of phage surface receptors

The infection cycle of most tailed phages begins with host recognition by reversible adsorption to a first "primary" receptor on host cells (often a sugar motif on surface glycans) followed by irreversible binding to a terminal or "secondary" receptor directly on the cell surface, which results in DNA injection [33]. Importantly, the adsorption to many potential hosts is blocked by long O-antigen chains on the lipopolysaccharide (LPS) and other exopolysaccharides that effectively shield the cell surface unless they can be degraded or specifically serve as the phage's primary receptor [33–36]. Tailed phages bind to primary and secondary receptors using separate structures at the phage tail, which display the dedicated receptor-binding proteins (RBPs) in the form of tail fibers or central tail tips (comprehensively reviewed in [33]). Most commonly, surface glycans such as the highly variable O-antigen chains or the enterobacterial common antigen (ECA) are bound as primary receptors, and this interaction may or may not be required for infectivity. Secondary receptors on gram-negative hosts are near-exclusively porin-family outer membrane proteins for siphoviruses and core LPS sugar structures for podoviruses, while myoviruses were found to use either one of the two depending on phage subfamily or genus [33,37,38]. Notably, no host receptor is known for the vast majority of phages that have been studied, but understanding the genetic basis and molecular mechanisms underlying host recognition as the major determinant of phage host range is a crucial prerequisite for host range engineering or a rational application of phage therapy [39]. Since tail fibers and other host recognition modules are easily identified in phage genomes, we used phage receptor specificity as a model to demonstrate the usefulness of systematic phenotypic data with the BASEL collection as a key to unlock information hidden in the genome databases.

We therefore first experimentally determined the essential host receptor(s) of all phages in the BASEL collection before analyzing the phages' genomes for the mechanisms underlying receptor specificity. Briefly, the dependence on surface proteins was assessed by plating each phage on a set of more than 50 single-gene mutants of *E. coli* K-12 (S1 Table) or whole-genome sequencing of spontaneously resistant bacterial clones (see Materials and methods). The role of host surface glycans was quantified with *waaG* and *waaC* mutants that display different truncations of the LPS core, a *wbbL(+)* strain with restored O16-type O-antigen expression, and a *wecB* mutant that is specifically deficient in production of the ECA [40–42] (Fig 2A, S1 Table, and Materials and methods).

## Overview: Phenotyping of sensitivity/resistance to bacterial immunity systems

Each bacterial strain encodes a unique repertoire of a few very common and a larger number of rarer, strain-specific immunity systems [11], but it is unknown how far this diversity

impacts the isolation of phages or their efficacy in therapeutic applications. A systematic view on the sensitivity and resistance of different phage groups to different immunity systems might therefore enable us to select or engineer phages with a higher and more reliable potency for phage therapy or biotechnology. As an example, previous work showed that T-even phages or the Queuovirinae subfamily of Siphoviridae (*Seuratvirus* and *Nonagvirus* genera) exhibit broad resistance to RM systems due to the hypermodification of cytosines or guanosines in their genomes, respectively, which could be an interesting target for phage engineering [43,44]. However, it is unknown whether this mechanism of RM resistance (or any other viral anti-immunity function) actually results in a measurably broader phage host range.

We therefore systematically quantified the sensitivity of all phages of the BASEL collection against a panel of 11 different immunity systems and scored their infectivity on a range of pathogenic enterobacteria that are commonly used as model systems (see Materials and methods and Fig 2B). Briefly, we tested 6 RM systems by including each two type I, type II, and type III systems that differ in the molecular mechanisms of DNA modification and cleavage [10,45–47] (Fig 2B). Besides these, we included the most well-studied Abi systems of *E. coli*, RexAB of the lambda prophage, PifA of the F-plasmid, as well as the Old, Tin, and Fun/Z systems of the P2 prophage (Fig 2B). Previous work suggested that RexAB and PifA sense certain proteins of phages T4 and T7, respectively, to trigger host cell death by membrane depolarization [10]. Conversely, Old possibly senses the inhibition of RecBCD/ExoV during lambda infections and might kill by tRNA degradation, Fun/Z abolishes infections by phage T5 via an unknown mechanism, and Tin poisons the replicative ssDNA binding protein of T-even phages [48]. Beyond *E. coli* K-12, we quantified the infectivity of the BASEL collection on uropathogenic *E. coli* (UPEC) strains UTI89 and CFT073, enteroaggregative *E. coli* (EAEC) strain 55989, alternative laboratory strain *E. coli* B REL606, and *Salmonella enterica* subsp. *enterica* serovar Typhimurium strains 12023s and SL1344 (see Materials and methods and S1 Table).

## Properties of the Drexlerviridae family

Phages of the Drexlerviridae family (previously also known as T1 superfamily [28]) are small siphoviruses with genome sizes of ca. 43 to 52 kb (Fig 3A–3E). The BASEL collection contains 13 new Drexlerviridae isolates that are broadly spread out across the various subfamilies and genera of this family (Fig 3D). Though it has never been directly demonstrated, it seems likely that the Drexlerviridae use their lateral tail fibers in the same way as their larger cousins of the Markadamsvirinae (see below) to contact specific O-antigen glycans as primary receptors without depending on this interaction for host recognition [49] (Fig 3A). Consistently, the locus encoding the lateral tail fibers is very diverse among Drexlerviridae, and the proteins forming these tail fibers are only highly related at the far N-terminus of the distal subunits where these are likely attached to the virion (S1C Fig). Among the Drexlerviridae that we isolated, only JakobBernoulli (Bas07) shows robust plaque formation on *E. coli* K-12 MG1655 with restored O16-type O-antigen expression, suggesting that its lateral tail fibers can use this O-antigen as primary receptor (Fig 3A and 3E).

Just like the much larger Markadamsvirinae siphoviruses (see below), Drexlerviridae phages use a small set of outer membrane porins as their secondary/final receptors for irreversible adsorption and DNA injection. To the best of our knowledge, previous work had only identified the receptors of T1 and IME18 (FhuA), IME347 (YncD), IME253 (FepA), and of LL5 as well as TLS (TolC), while for an additional phage, RTP, no protein receptor could be identified [38,50–52]. Using available single-gene mutants, we readily determined the terminal receptor of 11 out of our 13 Drexlerviridae phages as FhuA, BtuB, YncD, and TolC (Fig 3C and 3D) but failed for 2 others, AugustePiccard (Bas01) and JeanPiccard (Bas02). However, whole-genome

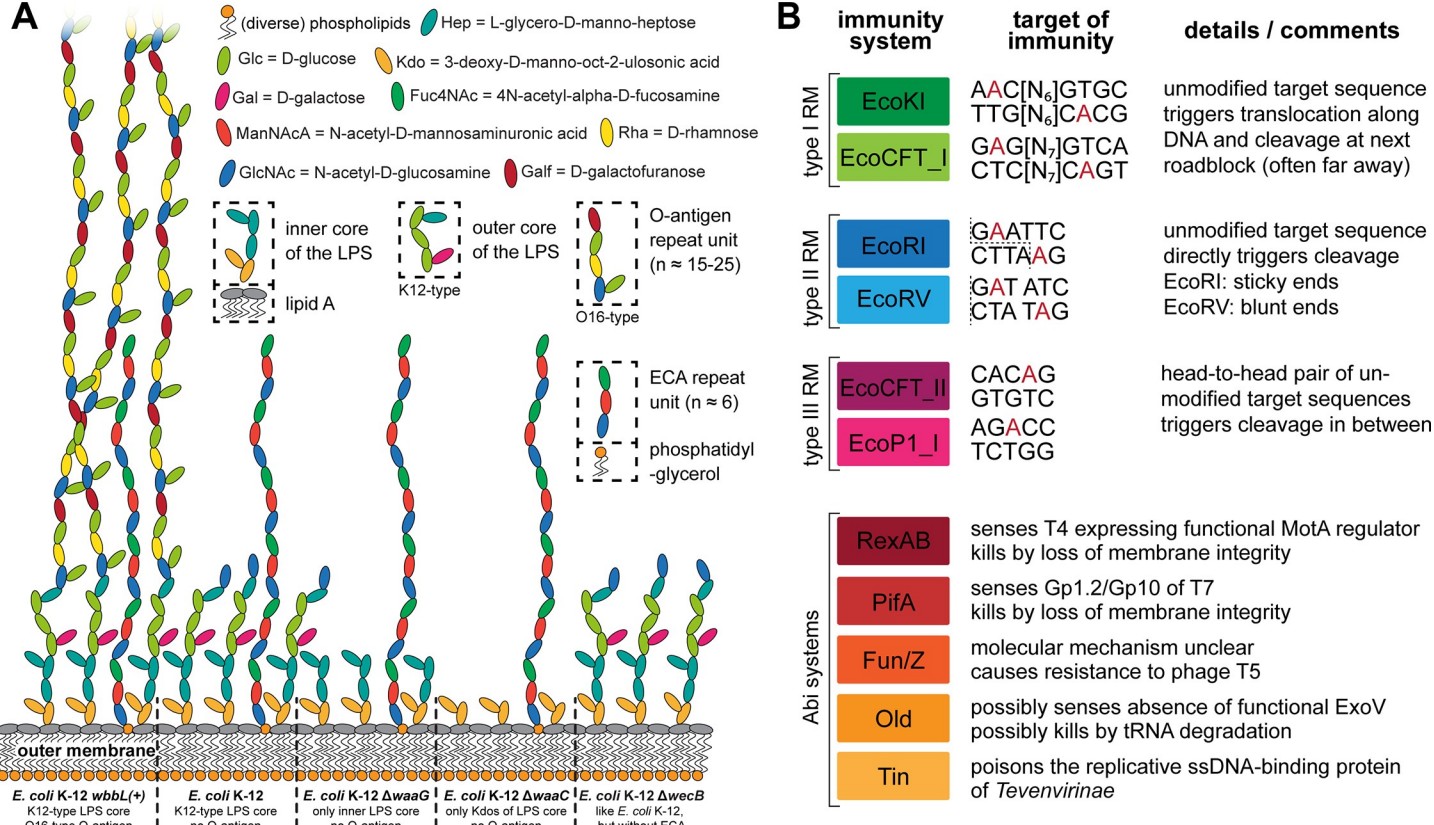

**Fig 2. Overview of *E. coli* surface glycan variants and the immunity systems used in this study. (A)** The surface glycans of different *E. coli* K-12 MG1655 variants are shown schematically (details in running text and Materials and methods). Note that the *E. coli* K-12 MG1655 laboratory wild type does not merely display the K12-type core LPS (classical rough LPS phenotype) but also the most proximal D-glucose of the O16-type O-antigen. **(B)** Key features of the 6 RM systems (each 2 of type I, type II, and type III) and the 5 Abi systems used for the phenotyping of this study are summarized schematically. Recognition sites of RM systems have either been determined experimentally or were predicted in REBASE (red nucleotides: methylation sites; dotted lines: cleavage sites) [45–47,129]. The Abi systems have been characterized to very different extent but constitute the most well-understood representatives of these immunity systems of *E. coli* [10,48]. Abi, abortive infection; ECA, enterobacterial common antigen; LPS, lipopolysaccharide; RM, restriction–modification.

sequencing of spontaneously resistant *E. coli* mutants showed that resistance was linked to mutations in the gene coding for LptD, the LPS export channel [34], strongly suggesting that this protein was the terminal receptor of these phages (Fig 4 and Materials and methods).

Similar to the central tail fibers of Markadamsvirinae, the RBPs of Drexlerviridae are thought to be displayed at the distal end of a tail tip protein related to well-studied GpJ of bacteriophage lambda [49,52,53]. The details of this host recognition module had remained elusive, though it was suggested that (like for T5 and unlike for lambda) dedicated RBPs are noncovalently attached to the J-like protein and might be encoded directly downstream of the *gpJ* homologs together with cognate superinfection exclusion proteins [52,53]. By comparing the genomes of all Drexlerviridae phages with experimentally determined surface protein receptors, we were able to match specific allelic variants of these bona fide RBP loci to each known surface receptor of this phage family (Fig 4A and 4B). Notably, for the 3 known phages targeting TolC including DanielBernoulli (Bas08), the RBP locus is absent and apparently functionally replaced by a C-terminal extension of the GpJ-like tail tip protein that probably directly mediates receptor specificity like GpJ of bacteriophage lambda (Fig 4B) [54,55]. Interestingly, similar RBP loci with homologous alleles are also found at the same genomic locus in small Siphoviridae of *Dhillonvirus*, *Nonagvirus*, and *Seuratvirus* genera (see below) where they

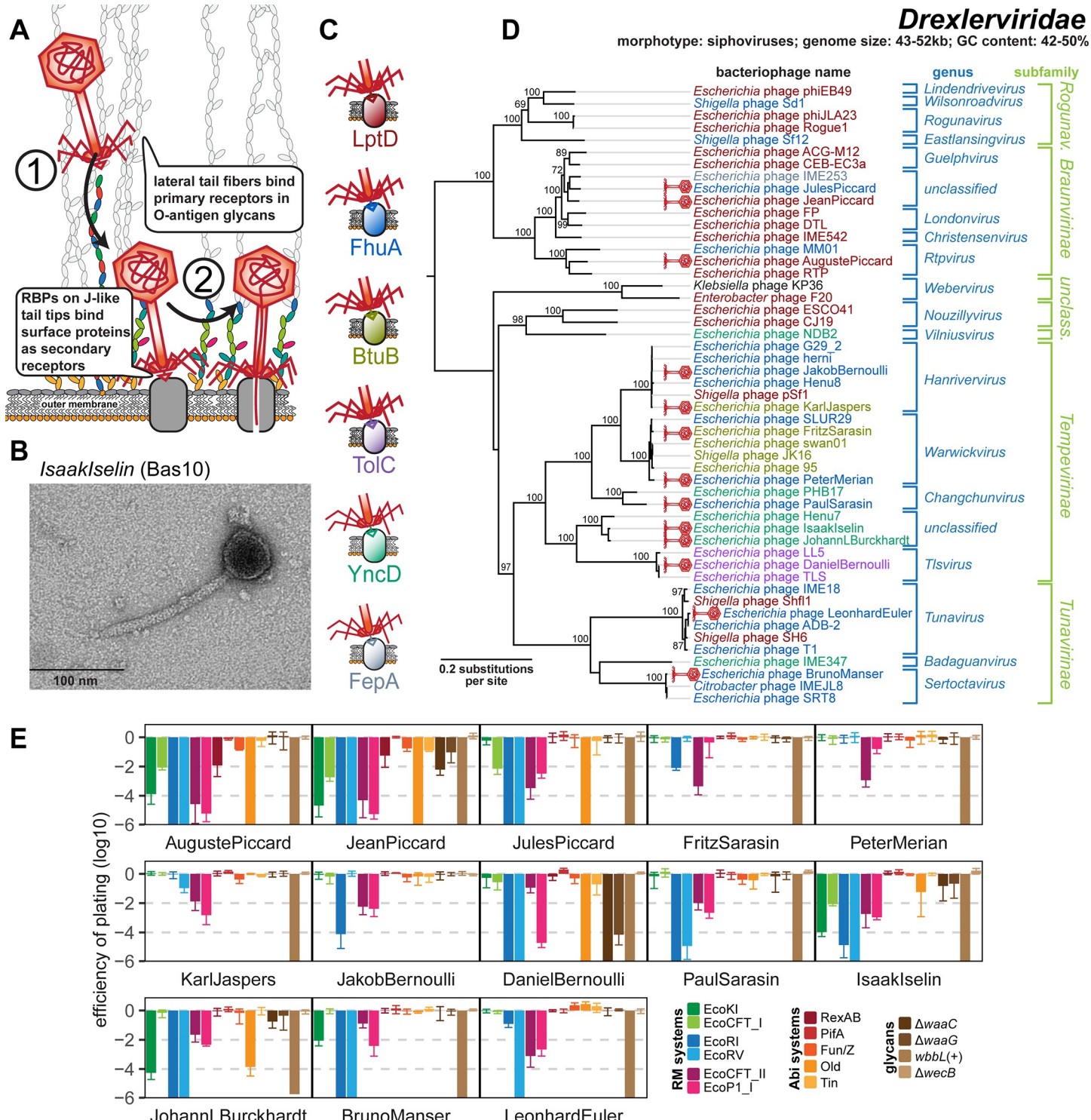

**Fig 3. Overview of Drexlerviridae phages.** (A) Schematic illustration of host recognition by Drexlerviridae. (B) Representative TEM micrograph of phage IsaakIselin (Bas10). (C) Color code of terminal receptor specificity. (D) Maximum-Likelihood phylogeny of Drexlerviridae based on several core genes with bootstrap support of branches shown if >70/100. Newly isolated phages of the BASEL collection are highlighted by red phage icons, and the determined or proposed terminal receptor specificity is highlighted at the phage names using the color code highlighted in (C). The phylogeny was rooted based on a representative phylogeny including *Dhillonvirus* sequences as outgroup (S1A Fig). (E) The results of quantitative phenotyping experiments with Drexlerviridae phages regarding sensitivity to altered surface glycans and bacterial immunity systems are presented as EOP. Data points and error bars represent average and standard deviation of at least 3 independent experiments. Raw data and calculations are available in S1 Data. BASEL, BActeriophage SElection for your Laboratory; EOP, efficiency of plating; RBP, receptor-binding protein; TEM, transmission electron microscopy.

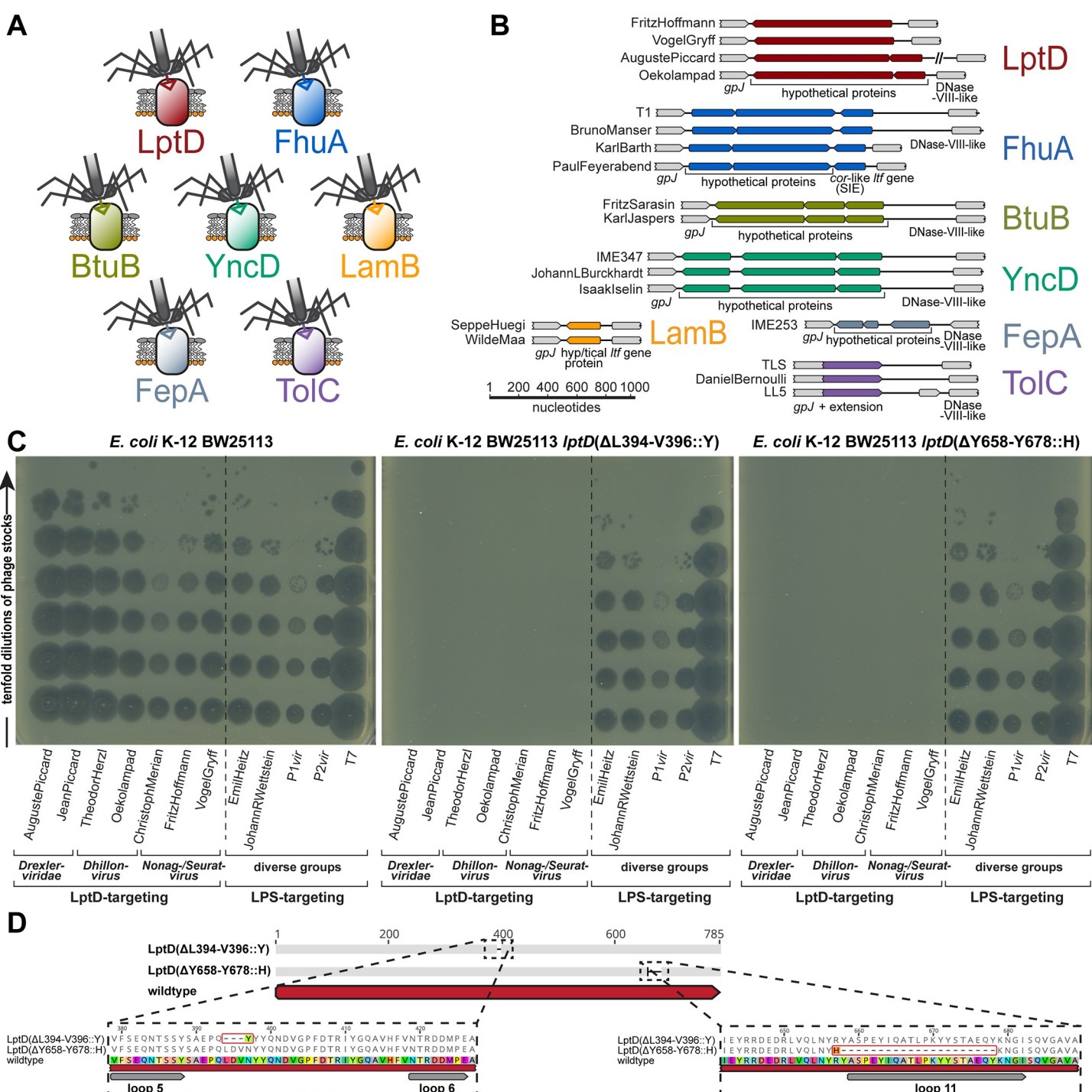

**Fig 4. Shared bona fide RBP loci of small siphoviruses mediate specificity to terminal receptors including LptD. (A)** The 7 identified receptors of small siphoviruses are shown with a color code that is also used to annotate demonstrated or predicted receptor specificity in the phylogenies of Figs 3D, 5D, and 5E. **(B)** We show representative bona fide RBP loci that seem to encode the receptor specificity of the small siphoviruses studied in this work (same color code as in (A)). Note that the loci linked to each receptor are very similar while the genetic arrangement differs considerably between loci linked to different terminal host receptors (see also S1B Fig). **(C)** Whole-genome sequencing of bacterial mutants exhibiting spontaneous resistance to 7 small siphoviruses with no previously known receptor revealed different mutations or small deletions in the essential gene *lptD* that encodes the LptD LPS export channel. Top agar assays with 2 representative mutants in comparison to the ancestral *E. coli* K-12 BW25113 strain were performed with serial 10-fold dilutions of 12 different phages (undiluted high-titer stocks at the bottom and increasingly diluted samples toward the top). Both mutants display complete resistance to the 7 small siphoviruses of diverse genera within Drexlerviridae and Siphoviridae families that share the same bona fide RBP module (S1B Fig) while no other phage of the BASEL collection was affected. We excluded indirect effects, e.g., via changes in the LPS composition in the *lptD* mutants, by confirming that 5 LPS-targeting phages of diverse families (see below) showed full infectivity on all strains. **(D)** The amino acid

sequence alignment of wild-type LptD with the 2 mutants highlighted in (A) shows that resistance to LptD-targeting phages is linked to small deletions in or adjacent to regions encoding extracellular loops as defined in previous work [150], suggesting that they abolish the RBP–receptor interaction. BASEL, BActeriophage SElection for your Laboratory; LPS, lipopolysaccharide; RBP, receptor-binding protein.

also match known receptor specificity without exception (Figs 4, 5, and S1B). Pending experimental validation, we therefore conclude that these distantly related groups of small siphoviruses share a common, limited repertoire of RBPs that enables receptor specificity of most representatives to be predicted in silico. As an example, it seems clear that phage RTP targets LptD as terminal receptor (Figs 3D and S1D). The poor correlation of predicted receptor specificity with the phylogenetic relationships of small siphoviruses (Figs 3D, 5D, and 5E) is indicative of frequent horizontal transfer of these RBP loci. This highlights the modular nature of this host recognition system, which might enable the targeted engineering of receptor specificity to generate "designer phages" for different applications as previously shown for siphoviruses infecting *Listeria* [14,56]. LptD would be a particularly attractive target for such engineering because it is strictly essential under all conditions, highly conserved, and heavily constrained due to multiple interactions that are critical for its functionality [34].

Once inside the host cell, Drexlerviridae phages are highly diverse in their sensitivity or resistance to diverse bacterial immunity systems (Fig 3E). While few representatives like Fritz-Sarasin (Bas04) or PeterMerian (Bas05) are highly resistant, most Drexlerviridae are very sensitive to any kind of RM systems sometimes including the type I machineries that are largely unable to inhibit any other phage that we tested (Fig 3E). Previous work on specific representatives suggested that the N6-adenine and/or C5-cytosine methyltransferases abundantly encoded by Drexlerviridae might counteract host restriction systems [22]. While this would be intuitive, we found no simple pattern linking methyltransferase genes or other genomic features to restriction resistance/sensitivity, suggesting that differences of this phenotype are driven by unknown factors or nuances in the activity or specificity of the DNA methyltransferases.

## Properties of Siphoviridae genera *Dhillonvirus*, *Nonagvirus*, and *Seuratvirus*

Phages of the *Dhillonvirus*, *Seuratvirus*, and *Nonagvirus* genera within the Siphoviridae family are small siphoviruses that are superficially similar to Drexlerviridae and have genomes with characteristic size ranges of 43 to 46 kb (*Dhillonvirus*), 56 to 61 kb (*Seuratvirus*), and 56 to 64 kb (*Nonagvirus*; see Fig 5A–5F). Our 12 isolates included in the BASEL collection are spread out broadly across the phylogenetic ranges of these genera (Fig 5D and 5E). Like Drexlerviridae (and Markadamsvirinae; see below), we suggest that they also recognize glycan motifs at the O-antigen as their primary receptor on different host strains (Fig 5A). This notion is strongly supported by the remarkable variation exhibited by the different genomes at the lateral tail fiber locus (S2A Fig), as observed previously [21,22]. None of these phages can infect *E. coli* K-12 MG1655 with restored O16-type O-antigen expression, but (like for Drexlerviridae) some require an intact LPS core for infectivity (Fig 5F). Experimental identification of the terminal receptor of all small Siphoviridae confirmed that receptor specificity seems to be encoded by the same system of bona fide RBP loci downstream of *gpJ* as for the Drexlerviridae (Figs 3C, 3D, 4A, 4B, and 5C–5E). Many of these phages target LptD or FhuA as terminal receptors, while others, unlike any Drexlerviridae, bind to LamB (Fig 5C–5E). Notably, 3 related *Nonagvirus* phages encode a distinct bona fide RBP module that could not be matched to any known terminal receptor (S2B Fig), suggesting that these phages are targeting a different protein.

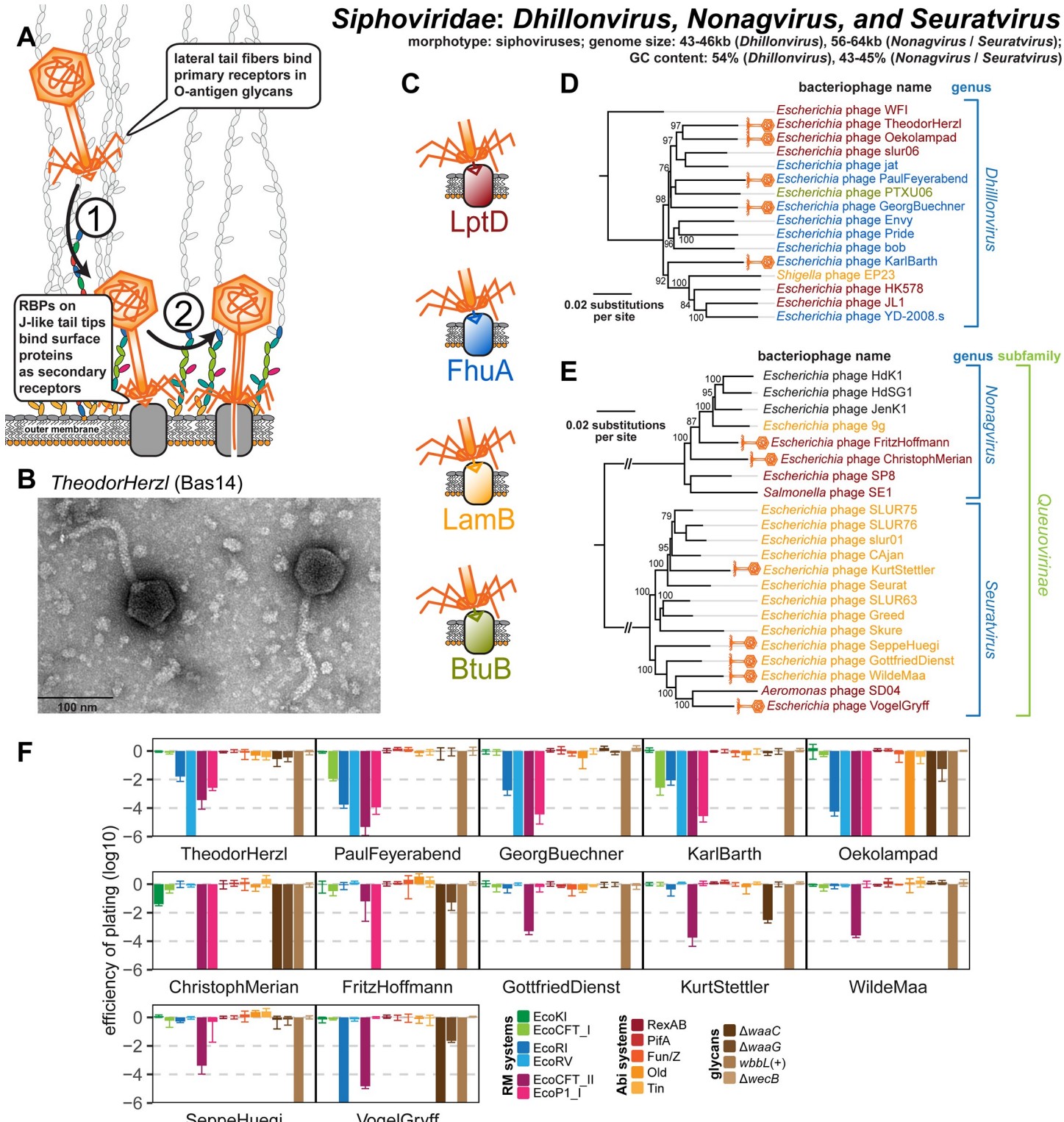

**Fig 5. Overview of Siphoviridae genera *Dhillonvirus*, *Nonagvirus*, and *Seuratvirus*. (A)** Schematic illustration of host recognition by small siphoviruses. **(B)** Representative TEM micrograph of phage TheodorHerzl (Bas14). **(C)** Color code of terminal receptor specificity (same as in Figs 3 and 4). **(D)** Maximum-Likelihood phylogeny of the *Dhillonvirus* genus based on a whole-genome alignment with bootstrap support of branches shown if >70/100. Newly isolated phages of the BASEL collection are highlighted by orange phage icons and the determined or proposed terminal receptor specificity is highlighted at the phage names using the color code highlighted in (C). The phylogeny was rooted between phage WFI and all others based on a representative phylogeny including Drexlerviridae sequences as outgroup

(S1A Fig). **(E)** Maximum-Likelihood phylogeny of the *Nonagvirus* and *Seuratvirus* genera based on a whole-genome alignment with bootstrap support of branches shown if >70/100. Newly isolated phages of the BASEL collection are highlighted by orange phage icons, and the determined or proposed terminal receptor specificity is highlighted at the phage names using the color code highlighted in (C) (see S2B Fig for the 3 phages without coloring). The phylogeny was rooted between the 2 genera that belong both to the Queuovirinae subfamily of Siphoviridae. **(F)** The results of quantitative phenotyping experiments with *Dhillonvirus*, *Nonagvirus*, and *Seuratvirus* phages regarding sensitivity to altered surface glycans and bacterial immunity systems are presented as EOP. Data points and error bars represent average and standard deviation of at least 3 independent experiments. Raw data and calculations are available in S1 Data. Abi, abortive infection; BASEL, BActeriophage SElection for your Laboratory; EOP, efficiency of plating; RBP, receptor-binding protein; RM, restriction–modification; TEM, transmission electron microscopy.

In difference to *Dhillonvirus* phages that are relatives of Drexlerviridae [21] (see also S1A Fig), *Nonagvirus* and *Seuratvirus* phages are closely related genera in the newly formed Queuovirinae subfamily with genomes that are 10 to 15 kb larger than those of the other small siphoviruses in the BASEL collection (S5 Table) [57,58]. This difference in genome size is largely due to genes encoding their signature feature, the 7-deazaguanine modification of $2'$-deoxy-guanosine (dG) in their genomes into $2'$-deoxy-7-cyano-7-deazaguanosine (dpreQ$_0$) for *Seuratvirus* phages and into $2'$-deoxyarchaosine (dG$^+$) for *Nonagvirus* phages [43] (S2C Fig). These modifications were shown to provide considerable (dG$^+$) or at least moderate (dpreQ$_0$) protection against restriction of genomic DNA of *Seuratvirus* phage CAjan and *Nonagvirus* phage *9g* in vitro, although chemical analyses showed that only around one-third (CAjan) or one-fourth (9g) of the genomic dG content is modified [43] (S2C Fig). Consistently, we found that Queuovirinae phages are remarkably resistant to type I and type II RM systems in our infection experiments, particularly if compared to other small siphoviruses of *Dhillonvirus* or Drexlerviridae groups (Figs 3E and 5F). Among *Nonagvirus* and *Seuratvirus* phages, only Christoph-Merian (Bas19, EcoKI) and VogelGryff (Bas25, EcoRI) show some sensitivity to these RM systems. However, all Queuovirinae phages are highly sensitive to type III RM systems (Fig 5F), possibly as a consequence of the much larger number of type III RM recognition sites in their genomes compared to those of the other types of RM systems (S5 Table; see also Fig 2B).

## Properties of Demerecviridae: Markadamsvirinae

Phages of the Markadamsvirinae subfamily of the Demerecviridae family (commonly known as T5-like phages [28]) are large siphoviruses with genomes of 101 to 116 kb length (Fig 6A–6E). Our 9 new isolates in the BASEL collection (plus well-studied phage T5) are well representative of the 2 major genera *Eseptimavirus* and *Tequintavirus* and their subclades (Fig 6C). These phages characteristically use their lateral tail fibers to bind each one or a few types of O-antigen very specifically as their primary host receptor, but this interaction is not essential for hosts like *E. coli* K-12 laboratory strains that do not express an O-antigen barrier [49] (Fig 6A). As observed previously [49,59], the diversity of O-antigen glycans is reflected in a high genetic diversity at the lateral tail fiber locus of the Markadamsvirinae of the BASEL collection (S3A Fig). Only one of these phages, IrisVonRoten (Bas32), can infect *E. coli* K-12 with restored expression of O16-type O-antigen, suggesting that it can recognize this glycan as its primary receptor (Fig 6E). However, several other isolates can infect different *E. coli* strains with smooth LPS or even *Salmonella*. Similar to what was proposed for the different small siphoviruses described above, the terminal receptor specificity of T5 and relatives is determined by dedicated RBPs attached noncovalently to the tip of a straight central tail fiber [49]. Compared to their smaller relatives, the repertoire of terminal receptors among Markadamsvirinae seems limited—the vast majority target BtuB (shown for, e.g., EPS7 [60] and S132 [61]), and each a few others bind FhuA (T5 itself [49] and probably S131 [61]) or FepA (H8 [62] and probably S124 [61]). Sequence analyses of the RBP locus allowed us to predict the terminal receptor of all remaining Markadamsvirinae, confirming that nearly all target BtuB (Figs 6C, 6D, and S3B).

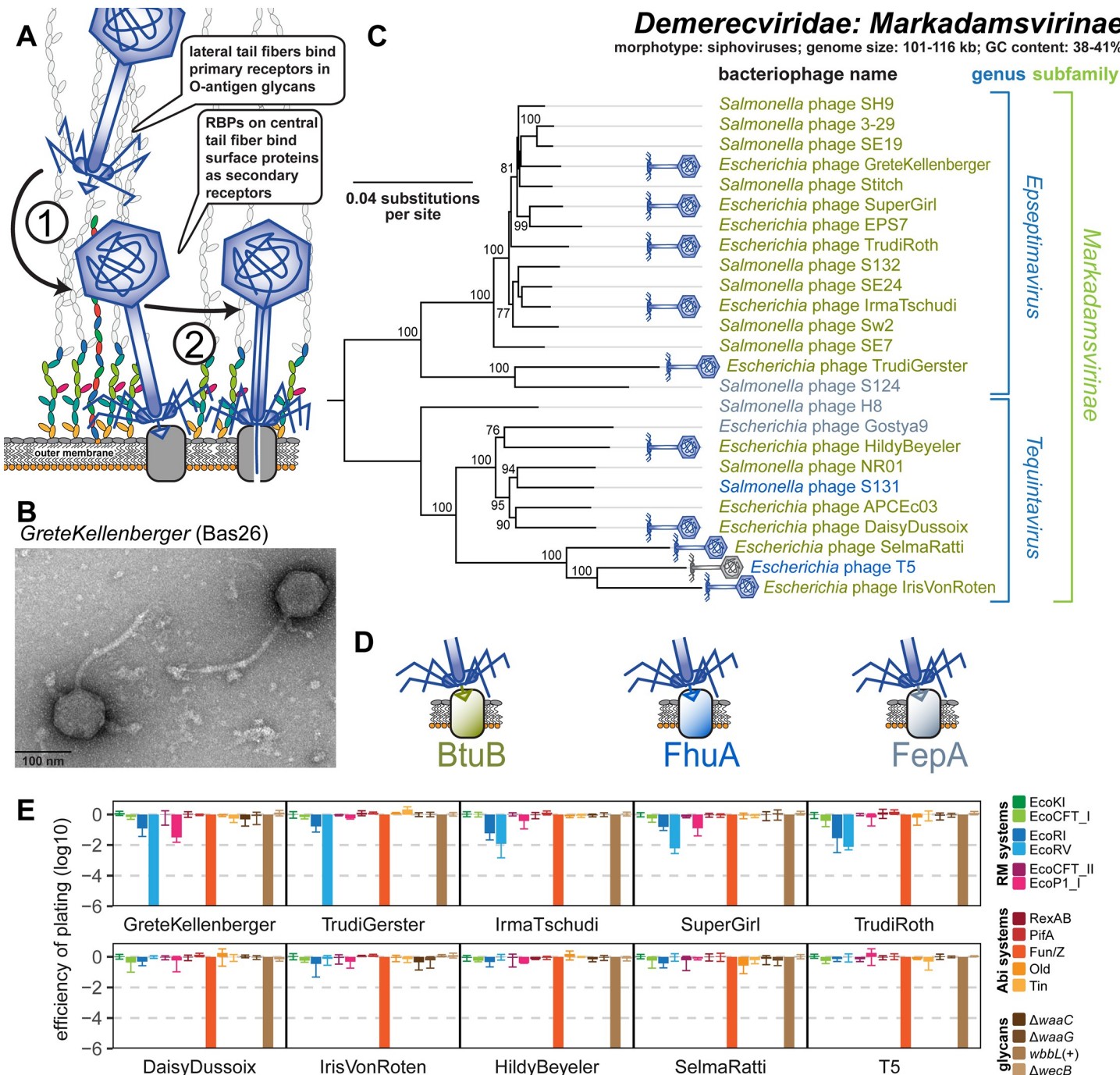

**Fig 6. Overview of Demerecviridae subfamily Markadamsvirinae.** (A) Schematic illustration of host recognition by T5-like siphoviruses. (B) Representative TEM micrograph of phage GreteKellenberger (Bas26). (C) Maximum-Likelihood phylogeny of the Markadamsvirinae subfamily of Demerecviridae based on several core genes with bootstrap support of branches shown if >70/100. Phages of the BASEL collection are highlighted by little phage icons, and the determined or proposed terminal receptor specificity is highlighted at the phage names using the color code highlighted in (D). The phylogeny was rooted between the *Epseptimavirus* and *Tequintavirus* genera. (D) Color code of terminal receptor specificity (same as in Figs 3–5). (E) The results of quantitative phenotyping experiments with Markadamsvirinae phages regarding sensitivity to altered surface glycans and bacterial immunity systems are presented as EOP. Data points and error bars represent average and standard deviation of at least 3 independent experiments. Raw data and calculations are available in S1 Data. Abi, abortive infection; BASEL, BActeriophage SElection for your Laboratory; EOP, efficiency of plating; RBP, receptor-binding protein; RM, restriction–modification; TEM, transmission electron microscopy.

Previous work showed that phage T5 is largely resistant to RM systems, and this was thought to be due to elusive DNA protection functions encoded in the early-injected genome region largely shared by T5 and other Markadamsvirinae [63]. Surprisingly, we find that this resistance to restriction is a shared feature only of the *Tequintavirus* genus, while their sister genus *Epseptimavirus* shows detectable yet variable sensitivity particularly to the type II RM system EcoRV (Fig 6E). It seems likely that this difference between the genera is due to the largely different number of EcoRV recognition sites in *Epseptimavirus* genomes (70 to 90 sites; S5 Table) and *Tequintavirus* genomes (4 to 18 sites; S5 Table). This observation suggests that efficient DNA ligation and RM site avoidance [64] play important roles in the restriction resistance of T5 and relatives besides their putative DNA protection system. Remarkably, all Markadamsvirinae are invariably sensitive to the Fun/Z Abi system as shown for T5 previously [48].

## Properties of Myoviridae: Tevenvirinae

Phages of the Tevenvirinae subfamily within the Myoviridae family are large myoviruses with characteristic prolate capsids and genomes of 160 to 172 kb size that infect a wide variety of gram-negative hosts [28] (Fig 7A–7E). The kinked lateral tail fibers of these phages contact primary receptors on the bacterial cell surface that are usually surface proteins like OmpC, Tsx, and FadL for the original "T-even" phages T4, T6, and T2, respectively, but can also be sugar motifs in the LPS like for T4 when OmpC is not available [65] (Fig 7A). Robust interaction with the primary receptor unpins the short tail fibers from the myovirus baseplate, which enables their irreversible adsorption to terminal receptors in the LPS core. Consequently, the tail sheath contracts like a syringe and the phage tail penetrates the cell envelope to eject the viral genome [33] (Fig 7A).

The 13 new T-even isolates in the BASEL collection mostly belong to different groups of the *Tequatrovirus* genus that also contains well-studied reference phages T2, T4, and T6, but 2 isolates were assigned to the distantly related *Mosigvirus* genus (Fig 7C). As expected, the infectivity of all tested Tevenvirinae shows strong dependence on each one of a small set of *E. coli* surface proteins that have previously been described as primary receptors of this group, i.e., Tsx, OmpC, OmpF, OmpA, and FadL [65] (Fig 7C and 7D). Interestingly, while for some of these phages the absence of the primary receptor totally abolished infectivity, others still showed detectable yet greatly reduced plaque formation (S4A Fig). It seems likely that these differences are caused by the ability of some Tevenvirinae lateral tail fibers to contact several primary receptors such as, e.g., OmpC and the truncated *E. coli* B LPS core in case of T4 [65,66].

Specificity for the secondary receptor depends on the short tail fibers that, in case of T4, target the lipid A–Kdo region deep in the enterobacterial LPS core [67]. Given that this region is still present in the *waaC* mutant, the most deep-rough mutant of *E. coli* K-12 that is viable (Fig 2A), it is unsurprising that T4 and some other Tevenvirinae did not seem to show a dependence on the LPS core in our experiments (Fig 7E). However, some *Tequatrovirus* isolates and all tested *Mosigvirus* isolates required an intact inner core of the host LPS for infectivity (Fig 7E). This phenotype is correlated with an alternative allele of the short tail fiber gene that varies between Tevenvirinae phages irrespective of their phylogenetic position (S4B Fig). We therefore suggest that those phages encoding the alternative allele express a short tail fiber that targets parts of the LPS core above the lipid A–Kdo region (S4B and S4C Fig; see also Fig 2A).

Besides receptor specificity, the phenotypes of our diverse Tevenvirinae phages were highly homogeneous. The hallmark of this Myoviridae subfamily is a high level of resistance to DNA-targeting immunity like RM systems because of cytosine hypermodification (hydroxymethyl-

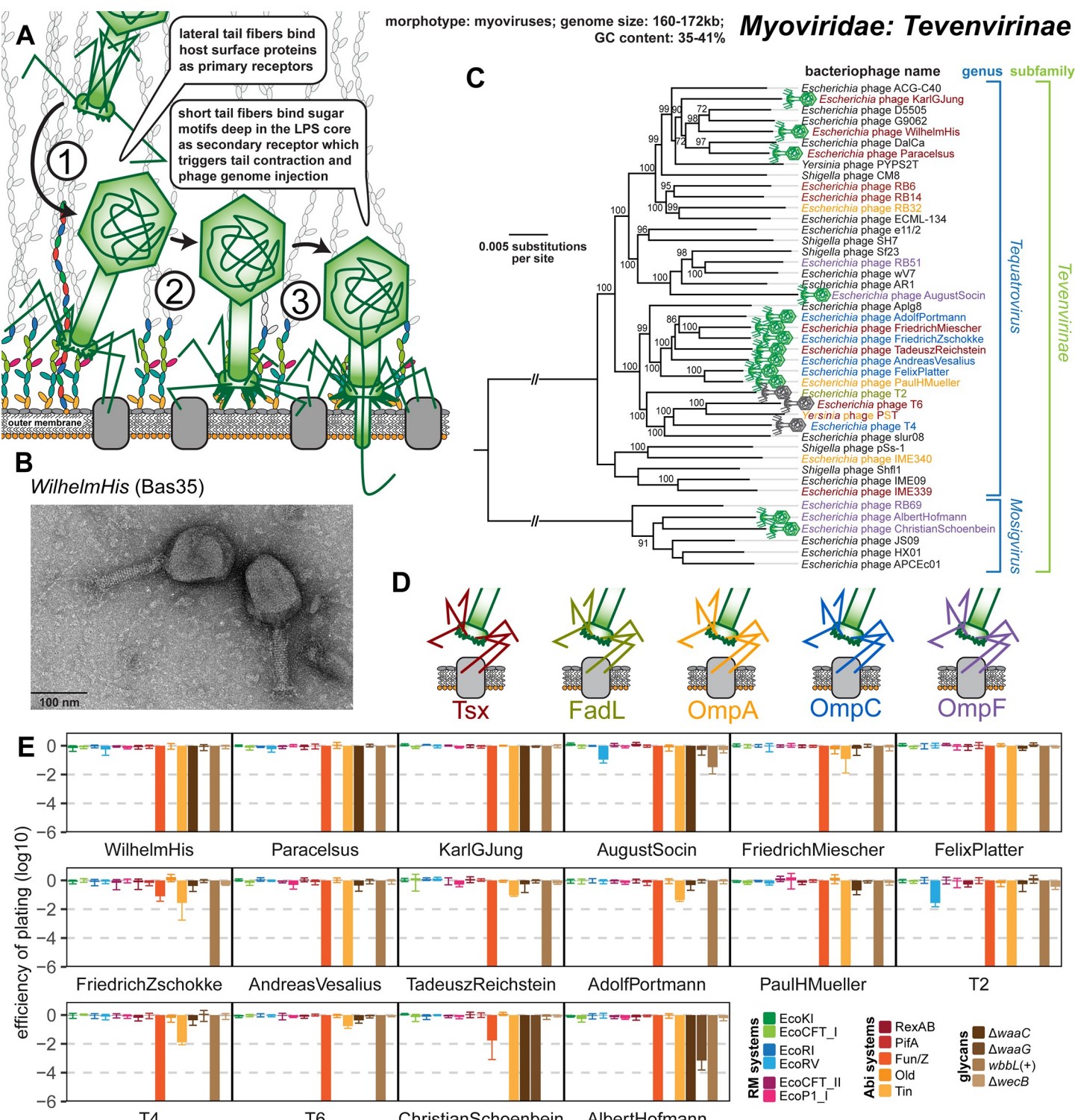

**Fig 7. Overview of the Myoviridae subfamily Tevenvirinae. (A)** Schematic illustration of host recognition by T-even myoviruses. **(B)** Representative TEM micrograph of phage WilhelmHis (Bas35). **(C)** Maximum-Likelihood phylogeny of the Tevenvirinae subfamily of Myoviridae based on a curated whole-genome alignment with bootstrap support of branches shown if >70/100. The phylogeny was rooted between the *Tequatrovirus* and *Mosigvirus* genera. Phages of the BASEL collection are highlighted by little phage icons and experimentally determined primary receptor specificity is highlighted at the phage names using the color code highlighted in (D). Primary receptor specificity of T-even phages depends on RBPs expressed either as a C-terminal extension of the distal half fiber (T4 and other OmpC-targeting phages) or as separate small fiber tip adhesins [65], but sequence analyses of the latter remained ambiguous. We therefore only annotated experimentally determined primary receptors (see also S4A Fig) [51,65]. **(D)** Color code of primary receptor specificity. **(E)** The results of quantitative phenotyping experiments with Tevenvirinae phages

regarding sensitivity to altered surface glycans and bacterial immunity systems are presented as EOP. Data points and error bars represent average and standard deviation of at least 3 independent experiments. Raw data and calculations are available in S1 Data. Abi, abortive infection; BASEL, BActeriophage SElection for your Laboratory; EOP, efficiency of plating; LPS, lipopolysaccharide; RBP, receptor-binding protein; RM, restriction–modification; TEM, transmission electron microscopy.

glucosylated for *Tequatrovirus* and hydroxymethyl-arabinosylated for *Mosigvirus*) [44,68]. Consistently, no or only very weak sensitivity to any of the 6 RM systems was detected for any tested T-even phage (Fig 7E). The weak sensitivity to EcoRV observed for AugustSocin (Bas38) and phage T2 does, unlike for Markadamsvirinae (see above), not correlate with a higher number of recognition sites and therefore possibly depends on another feature of these phages such as, e.g., differences in DNA ligase activity (S5 Table). Conversely, all tested T-even phages were sensitive to the Fun/Z and Tin Abi systems (Fig 7E). Tin was previously shown to specifically target the DNA replication of T-even phages [48], and we found that indeed all Tevenvirinae but no other tested phage are sensitive to this Abi system (Fig 7E). None of the tested T-even phages was sensitive to RexAB, the iconic Abi system targeting *rIIA/B* mutants of phage T4 [10], suggesting that RexAB is generally unable to inhibit wild-type T-even phages (Fig 7E; see S5A and S5B Fig for the RexAB sensitivity of a T4 *rIIA/B* mutant).

## Properties of Myoviridae: Vequintavirinae and relatives

Phages of the Vequintavirinae subfamily within the Myoviridae family are myoviruses with genomes of 131 to 140 kb size that characteristically encode 3 different sets of lateral tail fibers [21,69] (Fig 8A–8D). Besides Vequintavirinae of the *Vequintavirus* genus, this feature is shared by 2 groups of phages that are closely related to these Vequintavirinae sensu stricto: One group forms a cluster around phage phAPEC8 (genus *Phapecoctavirus* within Myoviridae), the other group is their sister clade including phage phi92 [70] and has so far remained unclassified (Fig 8C). Despite not being Vequintavirinae by current taxonomic classification, we are covering them together due to their considerable similarities and propose to classify the phi92-like phages (including 3 new isolates) as *Nonagintaduovirus* genus within the Vequintavirinae.

Three different, coexpressed lateral tail fibers have been directly visualized in the cryogenic electron microscopy structure of phi92, and large orthologous loci are found in all Vequintavirinae and relatives [21,69,70]. The considerable repertoire of glycan-hydrolyzing protein domains in these tail fiber proteins has been compared to a "nanosized Swiss army knife" and might be responsible for the exceptionally broad host range of these phages and their ability to infect even diverse capsulated strains of enterobacteria [21,50,69,70]. However, it is far from clear which genes code for which components of the different lateral tail fibers. Three large lateral tail fiber genes of Vequintavirinae exhibit clear allelic variation between genomes and might therefore encode proteins at the distal end of the 3 tail fibers that contact variable host structures (S5C Fig). However, most of the lateral tail fiber locus is highly conserved including multiple genes encoding proteins with sugar-binding or glycan-hydrolyzing domains (S5C Fig). This suggests that good part of these tail fibers might not have evolved to counter host variability but probably target conserved glycans on the host surface. Consistently, all tested Vequintavirinae and relatives seem to use the ECA as their shared primary receptor because their infectivity is compromised on a *wecB* knockout (Fig 8A and 8D). If true, the Vequintavirinae and relatives would be effectively monovalent on *E. coli* K-12 and might primarily use their "nanosized Swiss army knife" of tail fibers to bind and overcome capsules or other more specialized exopolysaccharides on diverse enterobacteria [69,70].

The terminal receptor of Vequintavirinae has been unraveled genetically for phage LL12, a close relative of rV5 (Fig 8C), and seems to be at the first, heptose-linked glucose of the LPS outer core, which is shared by all *E. coli* core LPS types [50,71] (Fig 8A). Consistently, we

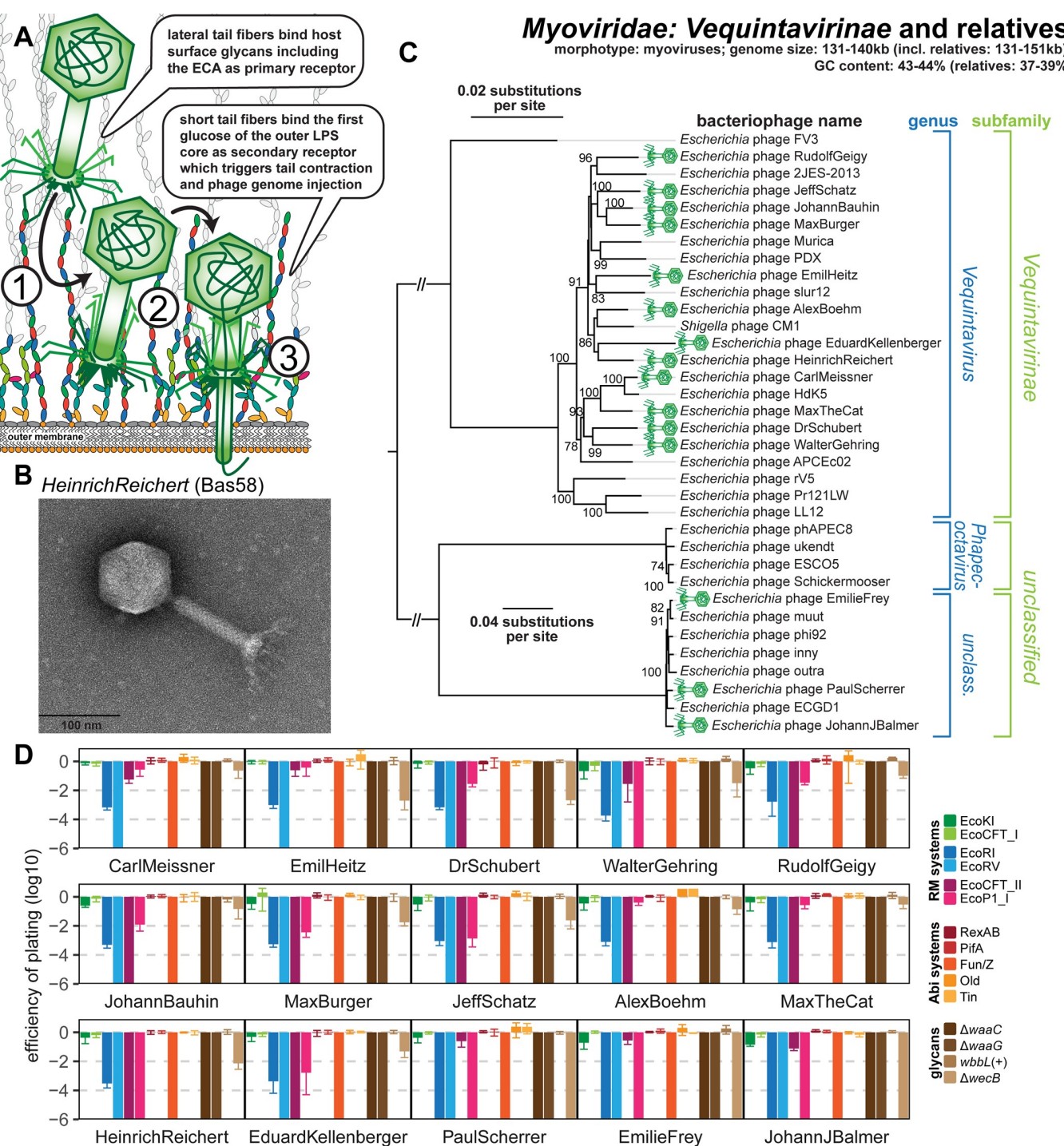

**Fig 8. Overview of the Myoviridae subfamily Vequintavirinae and relatives. (A)** Schematic illustration of host recognition by Vequintavirinae and related myoviruses. **(B)** Representative TEM micrograph of phage HeinrichReichert (Bas58). **(C)** Maximum-Likelihood phylogeny of the Vequintavirinae subfamily of Myoviridae and relatives based on a curated whole-genome alignment with bootstrap support of branches shown if >70/100. The phylogeny was rooted between the Vequintavirinae sensu stricto and the 2 closely related, formally unclassified groups at the bottom. Newly isolated phages of the BASEL collection are highlighted by green phage icons. **(D)** The results of quantitative phenotyping experiments with Vequintavirinae and their phi92-like relatives regarding sensitivity to altered surface glycans and bacterial immunity systems are presented as EOP. Data points and error bars represent average and standard deviation of at least 3 independent experiments. Raw data and calculations are available in S1 Data. Abi, abortive infection; BASEL, BActeriophage SElection for your Laboratory; ECA, enterobacterial common antigen; EOP, efficiency of plating; LPS, lipopolysaccharide; RM, restriction–modification; TEM, transmission electron microscopy.

found that all Vequintavirinae in the BASEL collection are completely unable to infect the *waaC* and *waaG* mutants with core LPS defects that result in the absence of this sugar (Fig 8D; see also Fig 2A). This observation would be compatible with the idea that all tested Vequinta-virinae including the phi92-like phages use the same secondary receptor. Indeed, the two very similar short tail fiber paralogs of Vequintavirinae sensu stricto do not show any considerable allelic variation between genomes (S5C Fig; unlike, e.g., the Tevenvirinae short tail fibers pre-sented in S4B and S4C Fig) and are also closely related to the single short tail fiber protein of phi92-like phages or *Phapecoctavirus* (S5D Fig).

Unlike large myoviruses of the Tevenvirinae subfamily, the Vequintavirinae and relatives are exceptionally susceptible RM systems, albeit with minor differences in the pattern of sensi-tivity and resistance from phage to phage (Fig 8D). At least under laboratory conditions, these phages therefore seem to lack any effective protection against this highly common type of bacterial immunity despite, e.g., a putative dTDP glycosylation machinery encoded by the phi92-like phages [22].

## Properties of Autographiviridae: Studiervirinae and Schitoviridae: Enquatrovirinae

Podoviruses of the Autographiviridae family like T3 and T7 or the Schitoviridae family like N4 have characteristic, stubby tails that contact the phages' terminal receptors on the cell surface for irreversible adsorption and DNA injection after initial host recognition by lateral tail fibers (Fig 9A–9F).

**Autographiviridae: Studiervirinae—T3, T7, and relatives.**   Phages T3 and T7 belong to the large Studiervirinae subfamily of Autographiviridae that typically carry genomes of ca. 37 to 41 kb in size and target LPS receptors on the host surface, which has been most well studied for iconic T7. The lateral tail fibers of this phage target the rough LPS of *E. coli* K-12 in ways that are not fully understood, possibly because several alternative and overlapping sugar motifs can be bound [72–74] (Fig 9A). Conformational changes in the stubby tail tube triggered by this receptor interaction then initiate the ejection of the phage genome from the virion [74,75]. Remarkably, at first, several internal virion proteins are ejected and then fold into an extended tail that spans the full bacterial cell envelope, which, in a second step, enables DNA injection into the host cytosol [76] (Fig 9A). Notably, our knowledge of this process does not allow the distinction between a "primary" receptor for host recognition and a "secondary" terminal receptor for irreversible adsorption and DNA injection. However, other Autographiviridae fol-low this classical scheme more closely and feature enzymatic domains at their tail fibers or tail spikes that likely mediate attachment to surface glycans as primary receptors followed by enzyme-guided movement toward the cell surface [36,77].

The Autographiviridae are named with reference to their single-subunit T3/T7-type RNA polymerase that plays key roles during phage infection but has also become a ubiquitous tool in biotechnology [78]. All Autographiviridae studied in this work belong to 3 different genera of the Studiervirinae subfamily that contain phages infecting diverse enterobacteria, suggesting frequent host switches (Fig 9D). Because phage T3 recognizes the truncated R1-type LPS core of *E. coli* B (see also below) and cannot infect K-12 strains, we generated a T3(K12) chimera that encodes the lateral tail fiber gene of T7 as was reported previously by others (see Materials and methods) [79]. As expected, all tested Autographiviridae use core LPS structures as host receptor and show impaired plaque formation on *waaC* and *waaG* mutants, but in almost all cases, some infectivity is retained even on the *waaC* mutant (Fig 9F). This suggests that, as pos-tulated for T7 [72–74], these phages are not strictly dependent on a single glycan motif but might recognize a broader range of target structures at the LPS core.

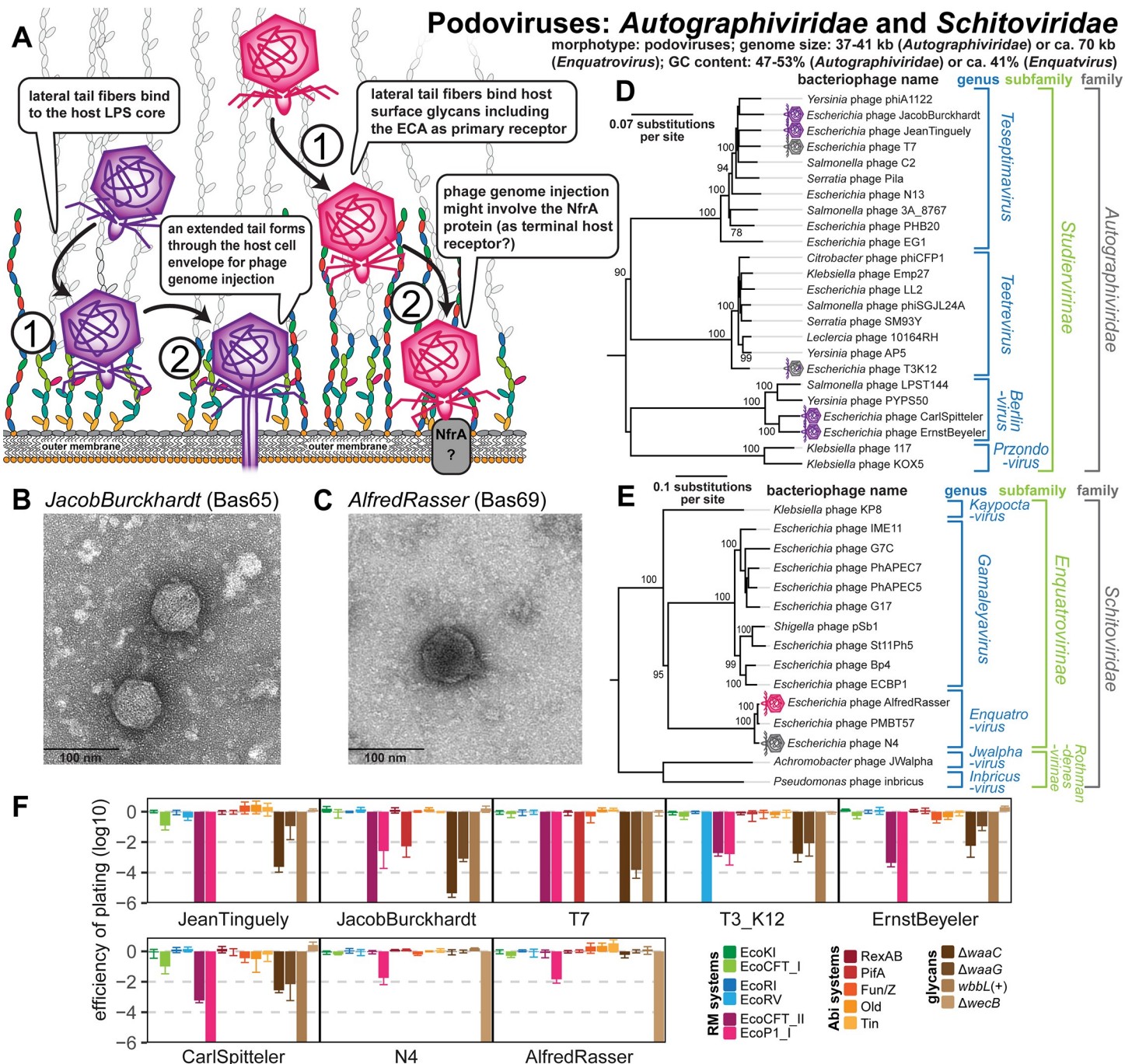

**Fig 9. Overview of Autographiviridae subfamily Studiervirinae and Schitoviridae subfamily Enquatrovirinae. (A)** Schematic illustration of host recognition by the studied Autographiviridae and Schitoviridae phages. **(B)** Representative TEM micrograph of T7-like Autographiviridae phage JacobBurckhardt (Bas65). **(C)** Representative TEM micrograph of *Enquatrovirus* AlfredRasser (Bas69). **(D)** Maximum-Likelihood phylogeny of the Studiervirinae subfamily of Autographiviridae based on several core genes with bootstrap support of branches shown if >70/100. The phylogeny was midpoint-rooted between the clade formed by *Teseptimavirus* and *Teetrevirus* and the other genera. Phages of the BASEL collection are highlighted by little phage icons. **(E)** Maximum-Likelihood phylogeny of the *Enquatrovirus* genus and related groups of Enquatrovirinae and other Schitoviridae based on several core genes with bootstrap support of branches shown if >70/100. The phylogeny was midpoint-rooted between the distantly related *Jwalphavirus* and *Inbricusvirus* genera and the other shown Schitoviridae. Phages of the BASEL collection are highlighted by little phage icons. **(F)** The results of quantitative phenotyping experiments with Autographiviridae and Schitoviridae phages regarding sensitivity to altered surface glycans and bacterial immunity systems are presented as EOP. Data points and error bars represent average and standard deviation of at least 3 independent experiments. Raw data and calculations are available in S1 Data. Abi, abortive infection; BASEL, BActeriophage SElection for your Laboratory; ECA, enterobacterial common antigen; EOP, efficiency of plating; LPS, lipopolysaccharide; RM, restriction–modification; TEM, transmission electron microscopy.

The overall pattern of restriction sensitivity and resistance is similar for all tested Autographiviridae. While type I and type II RM systems are largely ineffective, type III RM systems show remarkable potency across all phage isolates. Good part of the resistance to type II RM systems is likely due to the near-complete absence of EcoRI and EcoRV recognition sites in the genomes of these phages as described previously [64] (S5 Table) with the exception of 10 EcoRV sites in T3(K12), which, consequently, cause massive restriction (Fig 9F). The relatively few recognition sites for type I RM systems do not result in considerable sensitivity for any phage, because at the *gp0.3* locus, all of them either encode an Ocr-type DNA mimic type I restriction inhibitor like T7 (*Teseptimavirus*) or, like T3, an S-adenosylmethionine (SAM) hydrolase that deprives these RM systems of their substrate (all other genera) [9,80,81]. Notably, phages JacobBurckhardt (Bas65) and its close relative T7 are the only phages tested in our study that show any sensitivity to the PifA Abi system encoded on the *E. coli* K-12 F-plasmid (Fig 9F). Previous work showed that sensitivity of T7 to PifA immunity—as opposed to T3, which is resistant—depends on the dGTPase Gp1.2 of the phage, though the major capsid protein Gp10 also seems to play some role in sensitivity [82,83]. Consequently, we find that phages T7 and JacobBurckhardt, but not closely related *Teseptimavirus* JeanTinguely (Bas64), encode a distinct variant of dGTPase Gp1.2 that likely causes their sensitivity to PifA (S5E Fig).

**Schitoviridae: Enquatrovirinae—N4 and relatives.** Bacteriophage N4 and its close relative AlfredRasser (Bas69) are phages of the *Enquatrovirus* genus within the newly formed Schitoviridae family that typically carry genomes of ca. 68 to 74 kb and are hallmarked by a large, virion-encapsidated RNA polymerase used for the transcription of their early genes [31,58] (Fig 9A, 9C, and 9E). Based primarily on genetic evidence, phage N4 is thought to initiate infections by contacting the host's ECA with its lateral tail fibers [72,84,85] (Fig 9A). We indeed confirmed a remarkable dependence of N4 and AlfredRasser on *wecB* (Fig 9F) and detected homology between the glycan deacetylase domain of the N4 lateral tail fiber and a lateral tail fiber protein of Vequintavirinae, which also seem to use the ECA as their primary receptor (S5F Fig; see Fig 8). Like the O-antigen deacetylase tail fiber of its *Gamaleyavirus* relative G7C (Fig 9E), the enzymatic activity of the N4 tail fiber might provide a directional movement toward the cell surface that is essential for the next steps of infection [36,86] (Fig 9A). At the cell surface, the elusive outer membrane porin NfrA was suggested to be the terminal receptor of phage N4 based on the findings that it is required for N4 infection and interacts with the stubby tail of this phage [72,84,87]. To the best of our knowledge, this would make phage N4 the first podovirus with an outer membrane protein as terminal receptor [37,38], though the absence of homologs to the tail extension proteins of Autographiviridae indeed suggests a somewhat different mode of DNA injection (Fig 9A).

*Enquatrovirus* phages are highly resistant to any tested antiviral defenses with exception of a slight sensitivity to the EcoP1_I type III RM system (Fig 9F). While their resistance to tested type II RM systems is due to the absence of EcoRI or EcoRV recognition sites (S5 Table), the genetic basis for their only slight sensitivity to type III RM systems is not known. We note that *Enquatrovirus* phages encode an *rIIAB* locus homologous to the long-known yet poorly understood *rIIAB* locus found in many large myoviruses, which, in case of phage T4, provides resistance to the RexAB Abi system of phage lambda [88] (S5A and S5B Fig). Both phage N4 and AlfredRasser are indifferent to the presence of (O16-type) O-antigen or the tested truncations of the K-12 LPS core at the host cell surface (Fig 9F), suggesting that LPS structures play no role for their infection process.

## Properties of Myoviridae: Ounavirinae

The *Felixounavirus* genus in the Ounavirinae subfamily of Myoviridae comprises a group of phages with genomes of 84 to 91 kb and characteristically straight lateral tail fibers of which

*Salmonella* phage Felix O1 has been most well studied [28,89] (Fig 10A–10D). Previous work aimed at the isolation of *E. coli* phages differed greatly in the reported abundance of *Felixounavirus* phages, ranging from no detection [21] over a moderate number of isolates in most

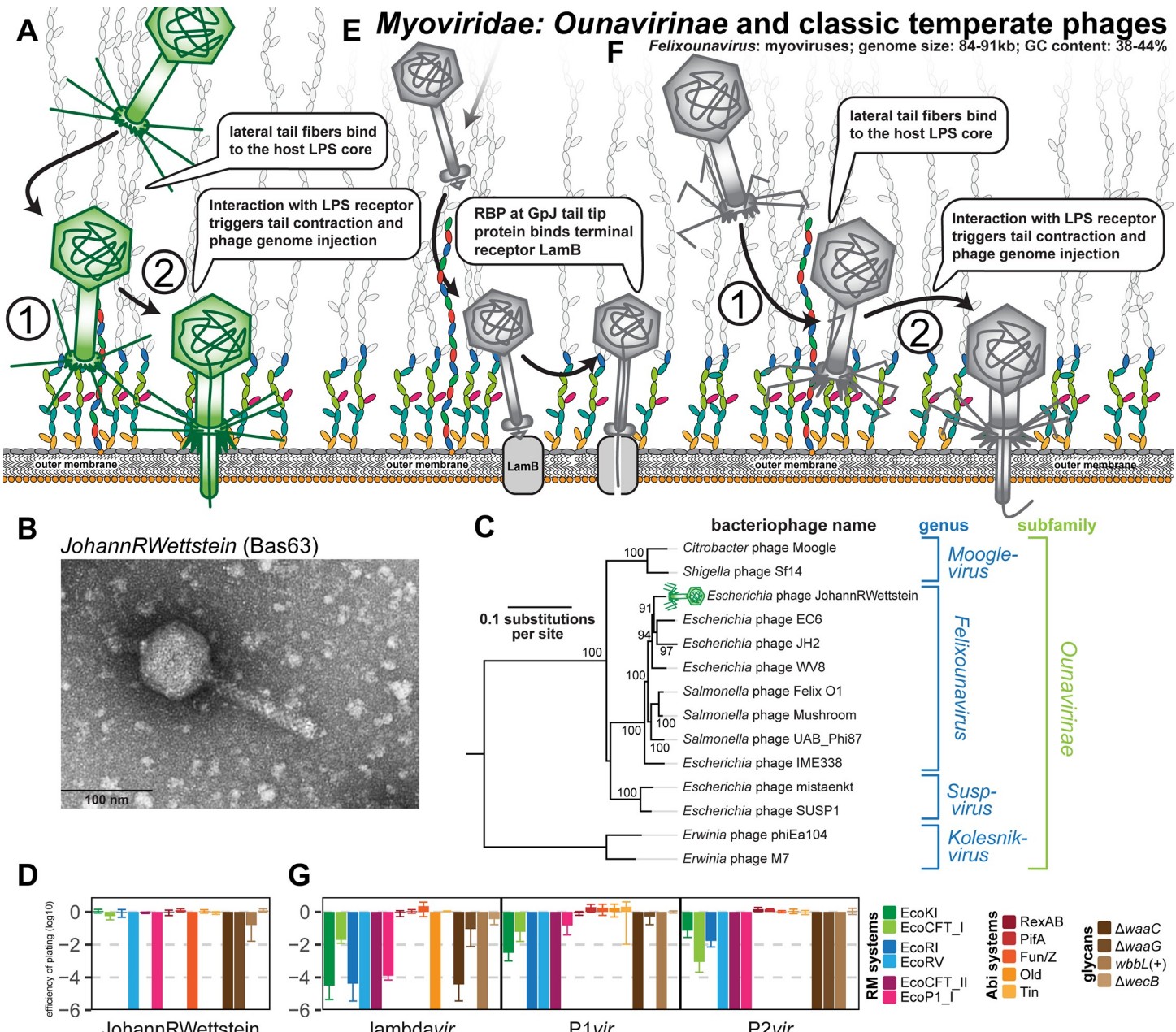

**Fig 10. Overview of Myoviridae: Ounavirinae and classic temperate phages. (A)** Schematic illustration of host recognition by Ounavirinae: *Felixounavirus* phages. Note that the illustration shows short tail fibers simply in analogy to Tevenvirinae or Vequintavirinae (Figs 7A and 8A), but any role for such structures has not been explored for Felix O1 and relatives. **(B)** Representative TEM micrograph of phage JohannRWettstein (Bas63). **(C)** Maximum-Likelihood phylogeny of the Ounavirinae subfamily of Myoviridae based on several core genes with bootstrap support of branches shown if >70/100. The phylogeny was midpoint-rooted between *Kolesnikvirus* and the other genera. Our new isolate JohannRWettstein is highlighted by a green phage icon. **(D)** The results of quantitative phenotyping experiments with JohannRWettstein regarding sensitivity to altered surface glycans and bacterial immunity systems are presented as EOP. **(E, F)** Schematic illustration of host recognition by classic temperate phages lambda, P1, and P2. Note the absence of lateral tail fibers due to a mutation in lambda PaPa laboratory strains [93]. **(G)** The results of quantitative phenotyping experiments with the 3 classic temperate phages regarding sensitivity to altered surface glycans and bacterial immunity systems are presented as EOP. Data points and error bars in (D) and (G) represent average and standard deviation of at least 3 independent experiments. Raw data and calculations are available in S1 Data. Abi, abortive infection; EOP, efficiency of plating; LPS, lipopolysaccharide; RBP, receptor-binding protein; RM, restriction–modification; TEM, transmission electron microscopy.

studies [23,90] to around one-fourth of all [22]. Across our phage sampling experiments, we found a single *Felixounavirus* phage, JohannRWettstein (Bas63), which is rather distantly related to Felix O1 within the eponymous genus (Fig 10C). Prototypic phage Felix O1 has been used for decades in *Salmonella* diagnostics because it lyses almost every *Salmonella* strain but only very few other Enterobacteriaceae (reviewed in [89]). The mechanism of its host recognition and the precise nature of its host receptor(s) have remained elusive, but it is generally known to target bacterial LPS and not any kind of protein receptors [89]. While many Ounavirinae seem to bind O-antigen glycans of smooth LPS, the isolation of multiple phages of this subfamily on *E. coli* K-12 with rough LPS and on *E. coli* B with an even further truncated LPS indicate that the functional expression of O-antigen is not generally required for their host recognition [22,89,91]. Phage JohannRWettstein is only slightly inhibited on *E. coli* K-12 with restored O16-type O-antigen, suggesting that it can either use or bypass these glycans and totally depends on an intact LPS core (Fig 10D). In addition, it is remarkably sensitive to several tested RM systems and shares sensitivity to the Fun/Z Abi system with all other tested Myoviridae and the Markadamsvirinae (Fig 10D).

## Properties of classical temperate phages lambda, P1, and P2

Besides the lytic T phages, temperate phages lambda, P1, and P2 have been extensively studied both as model systems for fundamental biology questions as well as regarding the intricacies of their infection cycle [7,32,48] (Fig 10E and 10F). Phage lambda was a prophage encoded in the original *E. coli* K-12 isolate and forms siphovirus particles that display the GpJ tail tip to contact the LamB porin as the terminal receptor for DNA injection [7,92], while the lateral tail fibers (thought to contact OmpA as primary receptor) are missing in most laboratory strains of this phage due to a mutation [7,93] (Fig 10E). We reproduced the dependence of our lambda*vir* variant on LamB and, as reported previously, found that an intact inner core of the LPS was required for infectivity ([72] and literature cited therein) (Fig 10G). Myoviruses P1 and P2 were both shown to require a rough LPS phenotype to contact their receptors in the LPS core, but the exact identity of these receptors has not been unraveled [38,48,72,94]. While the molecular details of the infection process after adsorption have not been well studied for P1, previous work showed that an interaction of the P2 lateral tail fibers with the LPS core of K-12 strains triggers penetration of the outer membrane by the tail tip [38,48,95] (Fig 10F). Consistently, our results confirm that mutations compromising the integrity of the K-12 core LPS abolished bacterial sensitivity to P1*vir* and P2*vir* [48,72,94] (Fig 10G). The quantification of these phages' sensitivity to different immunity systems revealed that they are remarkably sensitive to all tested RM systems (Fig 10G), a property only shared with a few Drexlerviridae (Fig 3E). As expected from the considerably lower number of restriction sites, the sensitivity was less pronounced for type I RM systems (Fig 10G and S5 Table). Phage lambda*vir* was additionally specifically sensitive to the Old Abi system of the P2 prophage (Fig 10G), as shown previously [48].

## Host range across pathogenic enterobacteria and laboratory wild-type *E. coli* B

The host range of bacteriophages is a critical feature for phage therapy because broad infectivity can enable phages to be used against different strains of the same pathogen without repeated sensitivity testing "analogous to the use of broad-spectrum antibiotics" [16]. Intuitively, across strains of the same host species, the infectivity of phages depends on their ability to successfully bind the variable surface structures of host cells and to overpower or evade strain-specific bacterial immunity [16,96] (Fig 11; see also Fig 2). While simple qualitative tests

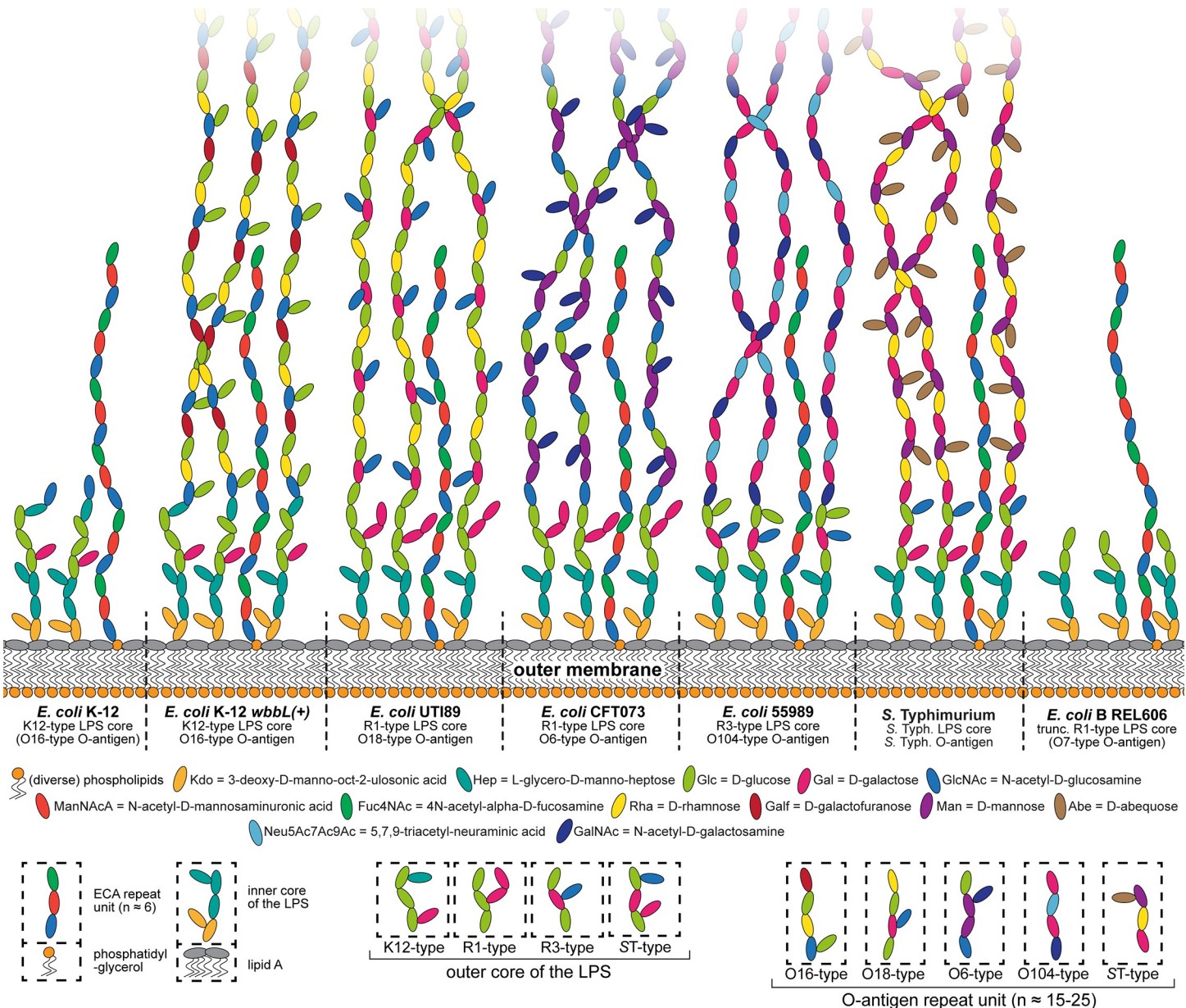

**Fig 11. Surface glycans of the enterobacterial strains used in this work.** The known surface glycans of different enterobacterial strains used in this work were drawn in comparison to highlight the diversity and barrier function of their O-antigen as well as the slight differences between LPS core types (see Materials and methods for details on how the illustration was composed). ECA, enterobacterial common antigen; LPS, lipopolysaccharide.

assessing the lysis host range primarily inform about host recognition alone, the more laborious detection of robust plaque formation as a sign of full infectivity is the gold standard of host range determination [16,96] (Fig 12A). We therefore challenged a panel of commonly used pathogenic enterobacterial strains with the BASEL collection and recorded the phages' lysis host range as well as plaque formation separately to gain insight into their host recognition as well as their ability to overcome immunity barriers inside host cells (Fig 12; see Materials and methods).

**Host range across pathogenic *E. coli* and *Salmonella* Typhimurium.** As expected from previous work, Vequintavirinae phages showed a broad lysis host range [21,23,69,70] and all

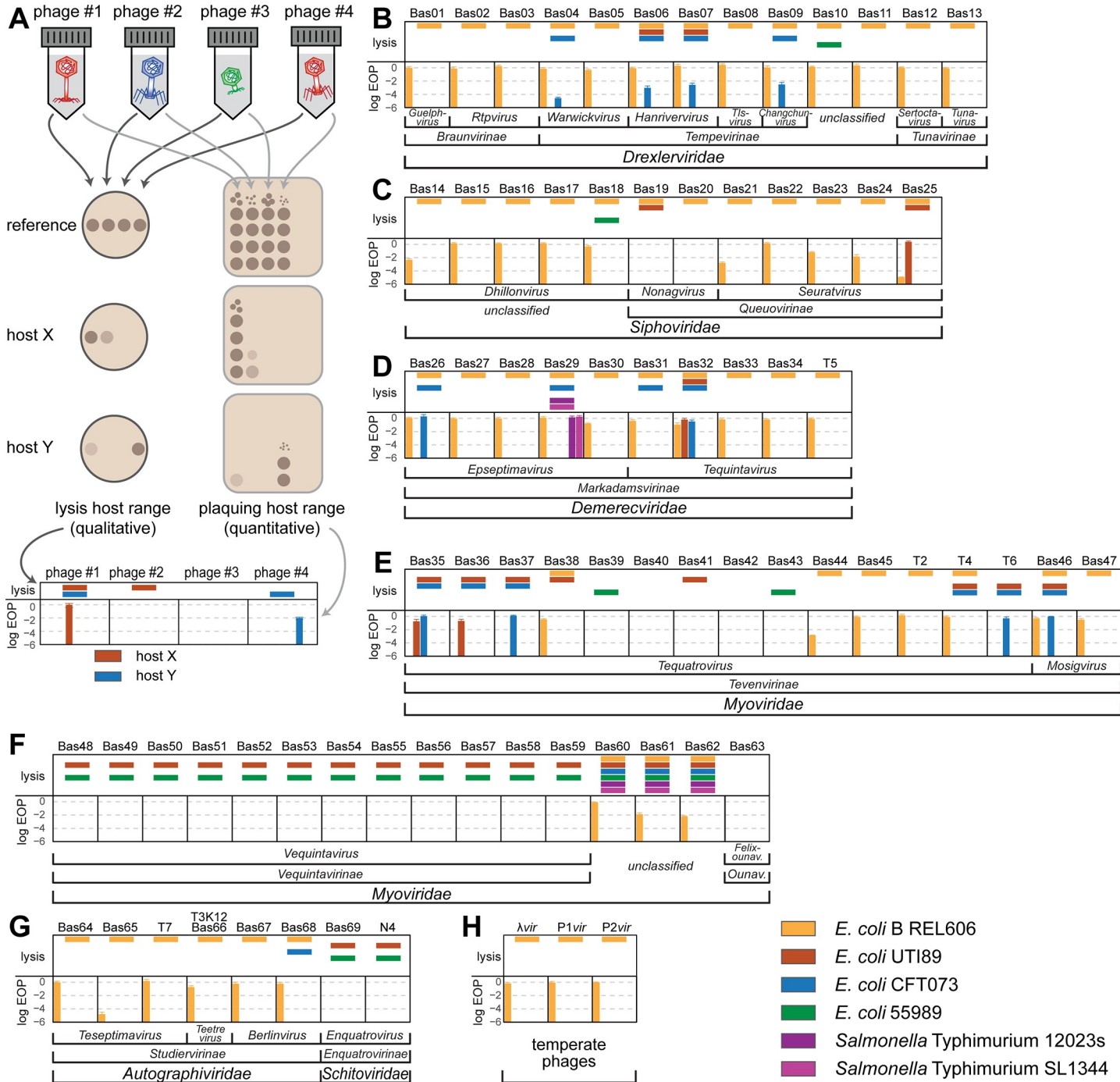

**Fig 12. Host range of phages in the BASEL collection. (A)** Schematic overview of qualitative spot test assays with high-titer phage stocks (>10⁹ pfu/ml; left) and quantitative EOP assays using serial dilutions of phage stocks (right) to determine the lysis host range and the plaquing host range of bacteriophages, respectively (see Materials and methods for details of the methodology). The graph at the bottom highlights how the results of the 4 fictional phages in this example would be plotted (with colorful bars at the top indicating the lysis host range and a bar diagrams for the EOP at the bottom that shows the plaquing host range). **(B-H)** The lysis host range and the plaquing host range of all phages in the BASEL collection across different enterobacteria were determined and visualized as shown in (A). Data points and error bars of the EOP plots represent average and standard deviation of at least 3 independent experiments. Raw data and calculations are available in S1 Data. BASEL, BActeriophage SElection for your Laboratory; EOP, efficiency of plating.

invariably infected *E. coli* UTI89, *E. coli* 55989, and *E. coli* K-12 with restored O16-type O-antigen (Figs 8D and 12F). This exact same lysis host range is shared by *Enquatrovirus* phages (Figs 9F and 12G), suggesting that this phenotype is driven by their shared recognition of the highly conserved ECA as primary receptor using a homologous RBP (S5F Fig). Recognition of the ECA molecules among smooth LPS by this glycan deacetylase tail fiber might enable Vequintavirinae and *Enquatrovirus* to bypass the O-antigen barrier and move to the cell surface along this glycan chain (see Fig 11) as previously described for *Salmonella* phage P22 and the N4 relative G7C that target O-antigen chains directly [36,86] (S5F Fig). Interestingly, the phi92-like relatives of Vequintavirinae have the broadest lysis host range of all tested phages and lysed all tested smooth *E. coli* and *Salmonella* hosts (Bas60-Bas62 in Fig 12G), consistent with previous impressive results for phi92 itself [70]. Given the remarkable lysis host range of all these bona fide ECA-targeting phages, it would be valuable to explore the molecular basis of their host recognition in future studies to use this knowledge for phage host range engineering [14].

Besides the bona fide ECA-targeting phages that can apparently bypass the O-antigen barrier, the lysis host range of only around a third of the other tested phages (24/61) included at least 1 host strain with smooth LPS, while the O16-type O-antigen of *E. coli* K-12 blocked infections by all but 5 of these phages scattered across the different taxonomic groups (5/61; Fig 12; see also Figs 3 and 5–10). In particular, the lysis and plaquing host ranges of tested siphoviruses were consistently narrow across taxonomic groups with a few exceptions especially among the T5-like phages (Markadamsvirinae; Fig 12B–12D). One of these isolates, SuperGirl (Bas29), is the only phage in the BASEL collection that shows robust plaque formation on *Salmonella* Typhimurium (Fig 12D). Taken together, these results confirm an important role of the O-antigen as a formidable barrier to bacteriophage infection that can only be overcome by specific recognition with tail fibers to breach it, e.g., via enzymatic activities [33].

A broad lysis host range was also observed for some T-even phages (Fig 12E), in line with previous work [21], though less homogeneous than for the bona fide ECA-targeting phages. For both groups of phages, it is astonishing how poorly their broad host recognition is reflected by actual plaque formation (Fig 12E–12G). Despite robust "lysis-from-without" in some cases [97], no reliable plaque formation on any other host than *E. coli* K-12 was observed for the bona fide ECA-targeting phages, and the range of plaque formation of Tevenvirinae was also severely contracted relative to the lysis host range (Fig 12E–12G). This observation highlights the potency of bacterial immunity systems as well as the well-known limitations of simple top agar spot tests to determine bacteriophage host range [16,96].

**Sensitivity and resistance of *E. coli* B to the BASEL collection phages.** The *E. coli* B lineage of laboratory strains are significant as main models of early bacteriophage research and as the original hosts of the T phages [5,98]. Like *E. coli* K-12, the B strains have lost O-antigen expression and even part of their R1-type LPS core during domestication [98]. The lack of this barrier enables most phages of the BASEL collection to lyse the typical *E. coli* B laboratory strain REL606 (Figs 11 and 12B–12H; see also S2 Text). Curiously, most bona fide ECA-targeting phages are unable to infect *E. coli* B (Fig 12F and 12G), possibly due to differences in ECA expression and/or the truncated R1-type LPS core. It seems likely that resistance of *E. coli* B to Vequintavirinae is the reason why no representative of this highly abundant group was sampled as one of the seven T phages. For different reasons, *E. coli* B strains also lack expression of *ompC*, and the REL606 strain additionally lacks functional *tsx* that both encode common primary receptors of T-even phages (Fig 7A–7D) [65,99]. Though phage T4 itself can use the *E. coli* B LPS core as a primary receptor if OmpC is absent [66], the inability of several other Tevenvirinae to infect *E. coli* B REL606 corresponds well to their dependency on OmpC and Tsx as primary receptors (compare Figs 7C and 12E). A diverse set of phages including, e.g.,

the *Nonagvirus* genus of small siphoviruses or the T7-like phage JacobBurckhardt (Bas65) can lyse *E. coli* B REL606 but show no or only poor plaque formation (Fig 12B–12H). Based on previous work, we suggest that this phenotype is caused by the specific repertoire of cryptic prophages of *E. coli* B strains that differ considerably from those of K-12 strains and are known to harbor active immunity systems [99].

## Discussion

### The BASEL collection as a representative and practical collection of *E. coli* phages

Assorted collections of model organisms like the BASEL collection exist, e.g., for *E. coli* in form of the ECOR collection of 72 *E. coli* strains that are commonly used to study how this species varies in certain traits or can deal with/evolves under certain conditions [100,101]. Nothing comparable had so far been available for bacteriophages, though the seven T phages had originally been chosen in the early 1940s with the explicit aim of providing a reference point that would enable comparative, systematic research on *E. coli* phages (reviewed in [6]). However, while these and the other classical model phages such as lambda have been invaluable to uncover many fundamental principles of molecular biology, their number and diversity was clearly too limited for a systematic view on bacteriophage biology. The BASEL collection closes this gap because it is broad enough for combined genomic and phenotypic studies aimed at uncovering molecular mechanisms of phage–host interactions as we show, e.g., for the terminal receptor specificity of small siphoviruses (Figs 3–5) and Tevenvirinae (S4B and S4C Fig). Diversity and size of the collection allowed us to uncover meaningful patterns of phage sensitivity or resistance to bacterial immunity systems within and across taxonomic groups (Figs 3 and 5–10). We envision that this approach will be helpful to understand how phages can overcome bacterial immunity (also, e.g., in clinical settings) but also for studies exploring the molecular biology of newly discovered bacterial immunity systems.

The composition of the BASEL collection mirrors the diversity of all common groups of *E. coli* phages as determined by previous work and also includes representatives of several rare groups to maximize the biological diversity, which could be probed for phage–host interactions (Fig 1; see Composition of the BASEL collection above). Any such collection of model organisms has inherent limitations that we discuss for the BASEL collection in S2 Text, but we feel that these do not affect the practical value of our phages. Most importantly, the absence of an O-antigen barrier from the *E. coli* K-12 MG1655 ΔRM sampling strain excluded any phages that depend on this glycan as an essential host receptor (like iconic *Salmonella* phage P22 [36]). To determine the impact of this limitation, we quantified the proportion of such O-antigen–dependent phages across sewage from 4 different treatment facilities and consistently found that O-antigen dependency is a very rare phenomenon (only a few percent of isolates; S6 Fig). Though it would be interesting to study the biology of these phages in more detail, their rarity and practical considerations (given that all used *E. coli* laboratory strains lack O-antigen) strongly suggest that their absence is not a major limitation of the BASEL collection.

### Remarkable patterns of bacteriophage receptor specificity

Our results regarding the terminal receptor specificity of different groups of siphoviruses inherently present the question why all these phages target less than 10 of the more than 150 outer membrane proteins of *E. coli* K-12 of which around 2 dozen are porins [37,38,102] (Figs 4A and 6D). Similarly, T-even phages use only 5 different outer membrane proteins as primary receptors (Fig 7D) [37,38]. Previous work suggested that this bias might be linked to the

abundance of the targeted proteins, both via a preference for particularly numerous proteins (favoring phage adsorption) or very scarce ones (avoiding competition among phages) [103]. However, we feel that this stark bias in phage preference might be largely driven by functional constraints. Notably, the large siphoviruses of Markadamsvirinae bind exactly a subset of those terminal receptors targeted by the small siphoviruses despite using nonhomologous RBPs, while there is no overlap between these proteins and the primary receptors of the Tevenvirinae [37,38] (Figs 3–7). It seems therefore likely that the receptors targeted by siphoviruses need to have certain properties, e.g., favoring DNA injection, that greatly limit the repertoire of suitable candidates. We envision that future studies will unravel the molecular mechanisms of how siphoviruses recognize and use outer membrane proteins as terminal receptors on gram-negative hosts similar to, e.g., how it has been studied for myoviruses and podoviruses that directly puncture the outer membrane [67,76]. Currently, it is even still unknown if the phage DNA is truly injected through the porin channel (as it is often drawn in the field [104] and also in our illustrations Figs 3A, 5A, and 6A) or rather passes the outer membrane adjacent to it through a phage-derived channel (as suggested by some experimental evidence (reviewed in [105])). Another open question amenable to combined genomic and phenotypic studies is whether the known specificity of siphoviruses for certain inner membrane channels (like ManYZ by phage lambda or PtsG for HK97 [7]) has a genetic basis similar to the proposed mix and match of RBPs and outer membrane proteins. Gaining a deeper understanding of bacteriophage host recognition would not only expand our knowledge of molecular mechanisms underlying bacteriophage ecology and evolution but could also help to apply this knowledge for biotechnology and therapeutic applications, e.g., in the context of "phage steering" [14,106].

## Bacteriophage sensitivity and resistance to host immunity

Our systematic analysis of the sensitivity and resistance profiles across the different taxonomic of phages of the BASEL collection revealed several strong patterns that are informative about underlying molecular mechanisms. Some groups of phages like Tevenvirinae or Queuovirinae are highly homogeneous in their profiles of sensitivity or resistance (Figs 5F and 7E), probably largely driven by conserved DNA modifications. Others are more heterogeneous, possibly due to a wide variety of known and unknown genetic factors such as restriction site avoidance or allelic differences that affect detection by Abi systems (see above, e.g., for Markadamsvirinae and Studiervirinae).

Overall, we observed that phages with bigger genomes such as Tevenvirinae, Vequintavirinae, and Markadamsvirinae are commonly targeted by Abi systems, while smaller siphoviruses or podoviruses are rarely sensitive to this type of bacterial immunity (Figs 3 and 5–10; see also S2 Text). Similarly, we confirmed the intuitive expectation that the potency of RM systems is linked to the number of recognition sites in each phage genome unless they are masked by DNA modifications. Consequently, large phages with many recognition sites (like Vequintavirinae and relatives) are exceptionally sensitive to RM systems (Fig 8D), while type I RM systems perform very poorly against most phages because their long recognition sites are rare (Fig 2B and S5 Table; see also Figs 3 and 5–10). For several groups of phages like Markadamsvirinae and some Tevenvirinae, the EcoRV system is the only RM system with significant impact (Figs 6E and 7E). We suggest that this might be due to the blunt-cutting activity of EcoRV that is much more difficult to seal for DNA repair than the sticky ends introduced by the other tested RM systems (Fig 2B), though phage DNA ligases can also differ in their efficiency on different kinds of DNA breaks [107].

One surprising result of our phenotyping was the remarkably poor target range of the most well-studied Abi systems of *E. coli* K-12, RexAB and PifA, that each affected only 2 or 3 phages

including all those previously known to be sensitive (Figs 3 and 5–10 as well as S5A and S5B Fig) [12]. Conversely, the three Abi systems of phage P2 protected each against a considerable number of phages, especially Fun/Z that inhibited all Tevenvirinae, Vequintavirinae, and Markadamsvirinae plus the *Felixounavirus* JohannRWettstein (Figs 3 and 5–11). One possible explanation could be that RexAB and PifA have actually not evolved to serve as bacterial immunity systems and instead primarily have other biological functions like the recently proposed role of RexAB in the lysogeny/lysis switch of phage lambda [108]. Temperate phages themselves seem generally to be highly sensitive to any kind of host immunity (Fig 10G), possibly because their evolution is less driven by selection to overcome host defenses but rather by optimizing the lysogens' fitness, e.g., by providing additional bacterial immunity systems [8,9].

Though the technicalities of our immunity phenotyping are subject to a few minor caveats (see Materials and methods and S2 Text), our results demonstrate how the BASEL collection can be used to explore the biology of bacterial immunity systems and the underlying molecular mechanisms. Several open questions remain: To which extent do the different layers of bacterial immunity limit bacteriophage host range? Why are some groups of phages like the Tevenvirinae so strongly targeted by seemingly unrelated Abi systems and other groups of phages apparently not at all? And how well would it be possible, based on systematic phenotypic and genomic data of phages and hosts, to predict the sensitivity of newly isolated phages to diverse immunity systems or even bacteriophage lysis and plaquing host ranges just from phage and host genome sequences?

## Royal families instead of Darwinian Demon: Trade-offs and ecological niches in phage evolution

Our results show that remarkable phage traits such as the broad host recognition of Vequintavirinae and their relatives or the massive resistance of T-even phages to RM systems do not generally confer these phages an exceptionally large plaquing host range (Figs 7E, 8D, 12E, and 12F). This observation could be caused by strong trade-offs between different phage fitness traits as proposed already in earlier work [109]. Unlike the hypothetical "Darwinian Demon" that can maximize all fitness traits simultaneously and would eventually dominate its ecosystem alone [110], the limits imposed by these trade-offs would instead drive the adaptation of different phage groups toward specific niches that enable their long-term coexistence [109]. In marine environments, the coexistence of a few specialized, highly successful phage groups was proposed to be stable in space and time because extinction of individual phages would usually result in their replacement by relatives from the same, highly adapted group [111]. This "royal family model" could elegantly explain why the same groups of *E. coli* phages sampled in our work for the BASEL collection have also been found again and again in previous studies that sampled in distant geographic locations and specific other environments [21–27] (Fig 1). Notably, the molecular mechanisms underlying a possible divergent adaptation of phage groups into "functional types" with different ecological strategies have largely remained elusive so far [112]. However, the results of our systematic study highlight clear and consistent phenotypic differences between the most common groups of *E. coli* phages and suggest that the BASEL collection could be a useful tool to explore the fundamental ecology of bacteriophages.

## Concluding remarks

A recent landmark study systematically explored the genetic profile of host requirements for several *E. coli* phages and uncovered multiple unexpected and exciting twists of phage biology such as the dependence of phage N4 on high levels of the second messenger cyclic di-GMP [72]. Our current study based on the BASEL collection represents a complementary approach

to explore the biology of *E. coli* phages not with elaborate genome-wide screens on the host side but rather by combining phenotypic and genomic analyses of a well-assorted set of bacteriophages. With this strategy, we achieved significant advances in bacteriophage biology regarding the recognition of host receptors, the sensitivity or resistance of phages to host immunity, and how these factors come together to determine bacteriophage host range. Our work therefore establishes the BASEL collection as a powerful tool to explore new aspects of bacteriophage biology by unraveling links between phage phenotypes and their genome sequences. Furthermore, our extensive characterization of the most abundant lineages of *E. coli* phages also provides a useful field guide for teaching and outreach activities analogous to the successful SEA-PHAGES initiative [113] and the diverse student and high school projects that have enabled our study (see Acknowledgments and S5 Table).

## Materials and methods

### Preparation of culture media and solutions

Lysogeny Broth (LB) was prepared by dissolving 10 g/l tryptone, 5 g/l yeast extract, and 10 g/l sodium chloride in Milli-Q $H_2O$ and sterilized by autoclaving. LB agar plates were prepared by supplementing LB medium with agar at 1.5% w/v before autoclaving. The M9RICH culture medium was conceived as a variant of regular M9 medium [114] supplemented with trace elements and 10% v/v LB medium (prepared without NaCl) to promote the growth of diverse enterobacterial strains. It was prepared from sterilized components by mixing (for 50 ml) 33.75 ml Milli-Q $H_2O$, 10 ml 5× M9 salts solution, 5 ml LB medium without NaCl, 500 μl 40% w/v D-glucose solution, 100 μl 1 M $MgSO_4$, and 5 μl 1 M $CaCl_2$ using sterile technique. Unless indicated otherwise, all components were sterilized by filtration (0.22 μm).

Phosphate-buffered saline (PBS) was prepared as a solution containing 8 g/l NaCl, 0.2 g/l KCl, 1.44 g/l $NA_2HPO_4$·$2H_2O$, and 0.24 g/l $KH_2PO_4$ with the pH adjusted to 7.4 using 10 M NaOH and sterilized by autoclaving. SM buffer was prepared as 0.1 M NaCl, 10 mM $MgSO_4$, and 0.05 M Tris (pH 7.5) using sterile technique.

### Bacterial handling and culturing

*E. coli* and *Salmonella* Typhimurium strains were routinely cultured in LB medium at 37°C in glass culture tubes or Erlenmeyer flasks with agitation at 170 rpm. For all phenotyping assays, the bacteria were instead grown in M9RICH, which supports robust growth of all strains to high cell densities. LB agar plates were routinely used as solid medium. Selection for genetic modifications or plasmid maintenance was performed with ampicillin at 50 μg/ml, kanamycin at 25 μg/ml, and zeocin at 50 μg/ml.

### Bacteriophage handling and culturing

Bacteriophages were generally cultured using the double-agar overlay method [115] with a top agar prepared as LB agar with only 0.5% w/v agar supplemented with 20 mM $MgSO_4$ and 5 mM $CaCl_2$. Top agar plates were incubated at 37°C, and plaques were counted as soon as they were identified by visual inspection. However, the plates were always incubated for at least 24 hours to record also slow-growing plaques. We routinely used the improvements of classical phage techniques published by Kauffman and Polz [116].

High-titer stocks of bacteriophages were generated using the plate overlay method. Briefly, top agar plates were set up to grow almost confluent plaques of a given phage and then covered with 12 ml of SM buffer. After careful agitation for 24 hours at 4°C, the suspension on each plate was pipetted off and centrifuged at 8,000 *g* for 10 minutes. Supernatants were sterilized

with few drops of chloroform and stored in the dark at 4˚C. For archiving, bacteriophages were stored as virocells at −80˚C [117].

## Bacterial strains and strain construction

All bacterial strains used in this work are listed in S1 Table, all oligonucleotide primers in S2 Table, and all plasmids in S3 Table. The identity of all strains of the KEIO collection used in this study was confirmed by diagnostic PCRs [118] (S7 Fig). The phenotypes of all bacterial mutants used in this study could be complemented using ectopic expression of the gene that was knocked out or mutated to exclude polar effects and other genetic artifacts (S8–S13 Figs; see technical details below at the section Plasmid construction).

*Escherichia coli* **K-12 MG1655 ΔRM.**   The standard laboratory strain *E. coli* K-12 MG1655 (CGSC #6300) was engineered into a more permissive host for bacteriophage isolation by knocking out the EcoKI type I RM system (encoded by *hsdRMS*) and the McrA, Mrr, and McrBC type IV restriction systems using lambda red recombineering [119] (see S1 Text and our previous work for more technical details [120]). The resulting strain lacks all known *E. coli* K-12 restriction systems and was therefore called *E. coli* K-12 ΔRM. To enable isolation of bacteriophages with tropism for the sex pilus of the F-plasmid conjugation system, we supplied *E. coli* K-12 MG1655 ΔRM with a variant of the F-plasmid in which the *pifA* immunity gene (which might otherwise have interfered with bacteriophage isolation) had been replaced with a zeocin resistance cassette by one-step recombineering (see S1 Text for details). The strain was additionally transformed with plasmid pBR322_ΔP*tet* of reference [121] as an empty vector control for experiments with plasmid-encoded immunity systems (see below).

*Escherichia coli* **K-12 MG1655 ΔRM mutants with altered surface glycans.**   To test the effect of altered surface glycans of bacteriophage infection, we generated derivatives of *E. coli* K-12 MG1655 ΔRM in which genes linked to specific glycans were knocked out. The *waaC* and *waaG* genes of the LPS core biosynthesis pathway were knocked out to generate mutants displaying a deep-rough or extremely deep-rough phenotype (Fig 2A; see also the considerations in S2 Text). Expression of the O16-type O-antigen of *E. coli* K-12 was restored by precisely removing the IS5 element disrupting *wbbL* [42] (Fig 2A). These mutants were generated by two-step recombineering (see details in S1 Text and a list of all strains in S1 Table). A strain specifically lacking the ECA was obtained as the *wecB* mutant of the KEIO collection [118]. This strain is deficient in WecB, the enzyme synthesizing UDP-ManNAcA (UDP-N-acetyl-D-mannosaminuronic acid), which is thought to be specifically required for the synthesis of the ECA but no other known glycan of *E. coli* K-12 [41].

*Escherichia coli* **K-12 BW25113 *btuB* and *tolC* knockout mutants.**   *btuB* and *tolC* knockout mutants isogenic with the KEIO collection were generated to use these strains lacking known bacteriophage receptors for qualitative top agar assays (see below). Details of the strain construction are provided in S1 Text.

**Other enterobacterial strains used for bacteriophage phenotyping.**   *E. coli* B REL606 is a commonly used laboratory strain with a parallel history of domestication to *E. coli* K-12 MG1655 [98]. *E. coli* UTI89 (ST95, O18:K1:H7) and *E. coli* CFT073 (ST73, O6:K2:H1; we used the variant with restored *rpoS* described previously [122]) are commonly used as model strains for UPEC and belong to phylogroup B2 [123]. *E. coli* 55989 (phylogroup B1, ST678, O104:H4) is commonly used as a model strain for EAEC and closely related to the Shiga toxin–producing *E. coli*, which caused the 2011 outbreak in Germany [124,125]. *Salmonella enterica* subsp. *enterica* serovar Typhimurium strains 12023s (also known as ATCC 14028) and SL1344 are both commonly used in laboratory experiments but exhibit phylogenetic and biological differences [126].

## Plasmid construction

Plasmid vectors were generally cloned following the method of Gibson and colleagues ("Gibson Assembly") [127] in which 2 or more linear fragments (usually PCR products) are ligated directionally guided by short 25 bp overlaps. Initially, some plasmids were also constructed using classical restriction-based molecular cloning. Briefly, a PCR-amplified insert and the vector backbone were each cut with appropriate restriction enzymes (New England Biolabs, Ipswich, Massachusetts, USA). After dephosphorylation of the backbone (using FastAP dephosphorylase; Thermo Scientific, Waltham, Massachusetts, USA), insert and backbone were ligated using T4 DNA ligase (Thermo Scientific). Local editing of plasmid sequences was performed by PCR with partially overlapping primers as described by Liu and Naismith [128]. *E. coli* strain EC100 *pir(+)* was used as host for all clonings. The successful construction of every plasmid was confirmed by Sanger Sequencing. A list of all plasmids used in this study is found in S3 Table, and their construction is summarized in S4 Table. The sequences of all oligonucleotide primers used in this study are listed in S2 Table.

For the series of plasmids encoding the 11 different bacterial immunity systems studied in this work (see Fig 2B), we used the EcoRI and EcoRV constructs of Pleška, Qian and colleagues [121] as templates and, consequently, the corresponding empty vector pBR322_ΔP*tet* as general cloning backbone and experimental control. The different immunity systems were generally cloned together with their own transcriptional promoter region. However, the *rexAB* genes are transcribed together with the *cI* repressor gene of the lambda prophage [7]. We therefore cloned them directly downstream of the P*tet* promoter of the pBR322 backbone and obtained a functional construct (validated in S5A Fig). The EcoKI, EcoRI, EcoRV, and EcoP1_I RM systems have been studied intensively in previous work [45–47]. Besides the type IA RM system EcoKI, we also cloned EcoCFT_I that is nearly identical to the well-characterized type IB RM system EcoAI but encoded in the genome of *E. coli* CFT073 that was available to us [45,129]. Similarly, the EcoCFT_II system was identified as a type III RM system in the *E. coli* CFT073 genome using REBASE [47,129]. Due to problems with toxicity, some immunity systems were cloned not into pBR322_ΔP*tet* but rather into a similar plasmid carrying a low-copy SC101 origin of replication (pAH186SC101e [122]; see S3 and S4 Tables as well as the considerations in S2 Text). Since we failed to obtain any functional construct for the ectopic expression of *pifA* (as evidenced by lack of immunity against T7 infection), we instead replaced the F(*pifA*::*zeoR*) plasmid in *E. coli* K-12 MG1655 ΔRM with a wild-type F plasmid that had merely been tagged with kanamycin resistance and encodes a functional *pifA* (pAH200e).

For the series of plasmids generated to complement the different *E. coli* mutants used in this study, the genes that were knocked out or mutated (*wecB*, *btuB*, *tolC*, *fhuA*, *yncD*, *lamB*, *tsx*, *fadL*, *ompA*, *ompC*, *ompF*, *lptD*, *waaC*, *waaG*) were each cloned downstream of the P*lac* promoter of low-copy plasmid pAH186SC101e [122] (see S3 Table for a list of plasmids and S4 Table for the technical details of plasmid construction). Genes were generally cloned with their native promoter as identified using RegulonDB ([130]; http://regulondb.ccg.unam.mx/) or simply including ca. 200 bp of upstream region for solitary genes when the promoter has remained undescribed (*yncD*). Four genes (*wecB*, *lamB*, *waaC*, *waaG*) are encoded in operons and were therefore cloned merely with their own bona fide ribosome-binding site by including ca. 20 to 25 bp upstream of the start codon. All plasmids complemented the phenotypes that we had observed for the mutant strains (S8–S13 Figs).

## Bacteriophage isolation

**Basic procedure.**   Bacteriophages were isolated from various different samples between July 2019 and March 2021 on *E. coli* K-12 MG1655 ΔRM carrying F(*pifA*::*zeoR*) as the host (see S5 Table for details) using a protocol similar to common procedures in the field [16].

Phage isolation was generally performed without an enrichment step to avoid biasing the isolation toward fast-growing phages (but see below).

For aqueous samples, we directly used 50 ml, while samples with major solid components (like soil or compost) were agitated overnight at 4°C in 50 ml of PBS to release viral particles. Subsequently, all samples were centrifuged at 8,000 $g$ for 15 minutes to spin down particles larger than viruses. The supernatants were sterilized treated with 2.5% v/v chloroform, which safely inactivates any bacteria as well as enveloped viruses but will generally leave most Caudovirales intact [16]. Subsequently, viral particles were precipitated by adding 1 ml of a 2-M ZnCl$_2$ solution per 50 ml of sample, mixing shortly by inversion, and incubating the suspension at 37°C without agitation for 15 minutes [131]. After precipitation, the samples were centrifuged again at 8,000 $g$ for 15 minutes, and the supernatant was discarded. The pellets were carefully resuspended in each 500 µl of SM buffer by agitation at 4°C for 15 minutes. Subsequently, the suspensions were cleared quickly using a tabletop spinner and mixed with 500 µl of bacterial overnight culture (resuspended in fresh LB medium to induce resuscitation). After incubation at room temperature for 15 minutes to promote phage adsorption, each mixture was added to 9 ml of top agar and poured onto a prewarmed square LB agar plate (ca. 12 cm × 12 cm). After solidification, the plates were incubated at 37°C for up to 24 hours.

**Isolation of bacteriophage clones.** Bacteriophages were visible as plaques forming in the dense bacterial growth of the top agar. For isolation of bacteriophage clones, they were picked from clearly separated plaques of diverse morphologies with sterile toothpicks and propagated at least 3 times via single plaques on top agars of the isolation host strain *E. coli* K-12 MG1655 ΔRM. To avoid isolating temperate phages or phages that are poorly adapted to *E. coli* hosts, we only picked clear plaques (indicative of virulent phages) and discarded isolates that showed poor plaque formation [16].

**Isolation of Autographiviridae using enrichment cultures.** The direct plating procedure outlined above never resulted in the isolation of phages belonging to the Autographiviridae like iconic T phages T3 and T7. Given that these phages are known for fast replication and high burst sizes [132], we therefore performed a series of enrichment culture isolation experiments to obtain phages forming the characteristically large, fast-growing plaques of Autographiviridae. For this purpose, we prepared M9 medium using a 5× M9 salts solution and chloroform-sterilized sewage plant inflow instead of water (i.e., containing ca. 40 ml of sewage plant inflow per 50 ml of medium) and supplemented it with 0.4% w/v D-glucose as carbon source. About 50 ml cultures were set up by inoculating these media with each 1 ml of an *E. coli* K-12 MG1655 ΔRM overnight culture and agitated the cultures at 37°C for 24 hours. Subsequently, the cultures were centrifuged at 8,000 $g$ for 15 minutes, and each 50 µl of supernatant was plated with the *E. coli* K-12 MG1655 ΔRM isolation strain in a top agar on one square LB agar plate. After incubation at 37°C for 3 or 4 hours, the first Autographiviridae plaques characteristically appeared (before most other plaques) and were picked and propagated as described above. Using this procedure, we isolated 4 different new Autographiviridae phages (see S5 Table and Fig 9D).

**Composition of the BASEL collection.** Bacteriophages were mostly isolated and characterized from randomly picked plaques in direct selection experiments, but we later adjusted the procedure to specifically isolate Autographiviridae, which were the only major group of phages previously shown to infect *E. coli* K-12 initially missing from our collection (see above). After every set of 10 to 20 phages that had been isolated, we performed whole-genome sequencing and preliminary phylogenetic analyses to keep an overview of the growing collection (see below). In total, more than 120 different bacteriophages were sequenced and analyzed of which we selected 68 tailed, lytic phage isolates to compose the BASEL collection (see Fig 1 and S5 Table for details). Phages closely related to other isolates were deliberately excluded

unless they displayed obvious phenotypic differences such as, e.g., a different host receptor. In addition to the 68 newly isolated bacteriophages, 10 classical model phages were included for genomic and phenotypic characterization, and we view these phages as an accessory part of the BASEL collection. These 10 phages were 6 of the 7 T phages (excluding T1 because it is a notorious laboratory contaminant [30]), phage N4, and obligately lytic mutants of the 3 well-studied temperate phages lambda, P1, and P2 [5,7,31,32] (Fig 1; see also S5 Table). To generate the T3(K12) chimera, the 3′ end of *gp17* lateral tail fiber gene of phage T7 was cloned into low-copy plasmid pUA139 with flanking regions exhibiting high sequence similarity to the phage T3 genome (generating pUA139_T7(*gp17*); see S4 Table). Phage T3 was grown on *E. coli* B REL606 transformed with this plasmid and then plated on *E. coli* K-12 MG1655 ΔRM to isolate recombinant clones. Successful exchange of the parental T3 *gp17* allele with the variant of phage T7 was confirmed by Sanger Sequencing.

All 68 newly isolated bacteriophages and the T3(K12) chimera are shared via the German Collection of Microorganisms and Cell Cultures (DSMZ) with DSM identifiers as listed in S5 Table.

## Qualitative top agar assays

The lysis host range of isolated bacteriophages on different enterobacterial hosts and their ability to infect strains of a set of KEIO collection mutants lacking each one surface protein (or isogenic mutants that were generated in this study; see S1 Table and S1 Text) were tested by qualitative top agar assays. For this purpose, top agars were prepared for each bacterial strain on LB agar plates. For regular round Petri dishes (ca. 9.4 cm diameter), we used 3 ml of top agar supplemented with 100 μl of bacterial overnight culture, while for larger square Petri dishes (ca. 12 cm × 12 cm), we used 9 ml of top agar supplemented with 200 μl of overnight culture. After solidification, each 2.5 μl of undiluted high-titer stocks of all tested bacteriophages ($>10^9$ pfu/ml) were spotted onto the top agar plates and dried into the top agar before incubation at 37°C for at least 24 hours. If lysis zones on any enterobacterial host besides our *E. coli* K-12 ΔRM reference strain were observed, we quantified phage infectivity in efficiency of plating (EOP) assays (see below). Whenever a phage failed to show lysis on a mutant strain lacking a well-known phage receptor, we interpreted this result as indicating that the phage infecting depends on this factor as host receptor.

## Efficiency of plating assays

The infectivity of a given bacteriophage on a given host was quantified by determining the EOP, i.e., by quantitatively comparing its plaque formation on a certain experimental host to plaque formation on reference strain *E. coli* K-12 MG1655 ΔRM carrying F(*pifA::zeoR*) and pBR322_ΔP*tet* [133]. Experimental host strains were identical to the reference strain with the difference that they either carried plasmids encoding a certain bacterial immunity system (S3 Table) or had a chromosomal modification changing surface glycan expression (Fig 2A and S1 Table).

For quantitative phenotyping, top agars were prepared for each bacterial strain on LB agar plates by overlaying them with top agar (LB agar containing only 0.5% agar and additionally 20 mM MgSO$_4$ as well as 5 mM CaCl$_2$; stored at 60°C) supplemented with a suitable bacterial inoculum. For regular round Petri dishes (ca. 9.4 cm diameter), we used 3 ml of top agar supplemented with 100 μl of bacterial overnight culture, while for larger square Petri dishes (ca. 12 cm × 12 cm), we used 9 ml of top agar supplemented with 200 μl of overnight culture. While the top agars were solidifying, serial dilutions of bacteriophage stocks (previously grown on *E. coli* K-12 MG1655 ΔRM to erase any EcoKI methylation) were prepared in sterile PBS.

Subsequently, each 2.5 μl of all serial dilutions were spotted on all top agar plates and dried into the top agar before incubation at 37°C for at least 24 hours. Plaque formation was recorded repeatedly throughout this time (starting after 3 hours of incubation for fast-growing phages). The EOP of a given phage on a certain host was determined by calculating the ratio of plaques obtained on this host over the number of plaques obtained on the reference strain *E. coli* K-12 MG1655 ΔRM carrying F(*pifA*::*zeoR*) and pBR322_ΔP*tet* [133]. Plaque counts and the calculation of EOP of all replicates of all experiments reported in this study (Figs 3E, 5F, 6E, 7E, 8D, 9F, 10D, 10G, and 12B–12H) as well as their summary statistics are compiled in S1 Data.

When no plaque formation could be unambiguously recorded by visual inspection, the EOP was determined to be below detection limit even if the top agar showed lysis from without (i.e., lysis zones caused by bacterial cell death without phage infection at an efficiency high enough to form plaques [97]). However, for all non-K12 strains of *E. coli* as well as *Salmonella* Typhimurium, we determined the lysis host range (i.e., the range of hosts on which lysis zones were observed) besides the numerical determination of EOP (see the illustration in Fig 12A). Occasionally, we found that certain phage/host pairs were on the edge between merely strong lysis from without and very poor plaque formation. Whenever in doubt, we recorded the result conservatively as an EOP below detection limit (e.g., for phage FriedrichMiescher on *E. coli* 55989, *Enquatrovirus* phages and EmilHeitz on *E. coli* UTI89, and all Vequintavirinae and relatives on the host expressing the Fun/Z Abi system).

## Bacteriophage genome sequencing and assembly

Genomic DNA of bacteriophages was prepared from high-titer stocks using the Norgen Biotek Phage DNA Isolation Kit according to the manufacturer's guidelines and sequenced at the Microbial Genome Sequencing Center (MiGS) using the Illumina NextSeq 550 platform. Trimmed sequencing reads were assembled using the Geneious Assembler implemented in Geneious Prime 2021.0.1 with a coverage of typically 50 to 100× (S5 Table). Usually, circular contigs (indicating a complete assembly due to the fusion of characteristically repeated sequence at the genome ends [134]) were easily obtained using the "Medium Sensitivity/Fast" setting. Consistently incomplete assemblies or local ambiguities were solved by PCR amplification using the high-fidelity polymerase Phusion (New England Biolabs) followed by Sanger Sequencing. For annotation and further analyses, sequences were linearized with the 5′ end set either to the first position of the small terminase subunit gene or the first position of the operon containing the small terminase subunit gene.

## Bacteriophage genome annotation

A preliminary, automated annotation of the genes in all genomes was generated using Multi-Phate [135] and then manually refined. For this purpose, whole-genome alignments of all new isolates within a given group of phages and well-studied and/or well-annotated references were generated using MAFFT v7.450 [136] implemented in Geneious 2021.0.1 and used to inform the annotation based on identified orthologs. Bona fide protein-coding genes without clear functional annotation were translated and analyzed using the blastp tool on the NCBI server (https://blast.ncbi.nlm.nih.gov/Blast.cgi), the InterPro protein domain signature database [137], as well as the Phyre2 fold recognition server ([138]; http://www.sbg.bio.ic.ac.uk/~phyre2/html/page.cgi?id=index). tRNA genes were predicted using tRNA-ScanSE ([139]; http://lowelab.ucsc.edu/tRNAscan-SE/), and spanins were annotated with help of the Spanin-DataBase tool [140]. While endolysin genes were easily recognized by homology to lysozyme-like proteins and other peptidoglycan hydrolases, holins were more difficult to identify when

no holins were annotated in closely related bacteriophage genomes. In these cases, we analyzed all small proteins (<250 amino acids) for the presence of transmembrane helices—a prerequisite for the functionality of known holins [141]—and studied their possible relationships to previously described holins using blastp and InterPro. In most but not all cases, bona fide holins encoded close to endolysin and/or spanin genes could be identified. The annotation of our genomes in a comparative genomics setup made it easily possible to identify the boundaries of introns associated with putative homing endonucleases and to precisely identify inteins (see also S5 Table). The annotated genome sequences of all 68 newly isolated phages are available at the NCBI GenBank database under accession numbers listed in S5 Table.

## Bacteriophage naming and taxonomy

Newly isolated bacteriophages were named according to rules and conventions in the field and classified in line with the rules of the International Committee on the Taxonomy of Viruses (ICTV) [58,142] (S5 Table). As a first step of taxonomic classification, phages were roughly sorted by family and genus based on whole-genome blastn searches against the nonredundant nucleotide collection database (https://blast.ncbi.nlm.nih.gov/Blast.cgi; see also [143]). For each such broad taxonomic group, we selected a set of reference sequences from NCBI GenBank that would cover the diversity of this group and that always included all members, which had already previously been studied intensively. Subsequently, we generated whole-genome alignments of these sets of sequences generated using MAFFT v7.450 [136] implemented in Geneious 2021.0.1. These alignments or merely regions encoding highly conserved marker genes such as the terminase subunits, portal protein gene, major capsid protein gene, or other suitable loci were then used to generate Maximum-Likelihood phylogenies to unravel the evolutionary relationships between the different bacteriophage isolates and database references (see S3 Text for details). Care was taken to avoid genome regions that are recognizably infested with homing endonucleases or that showed obvious signs of having recently moved by horizontal gene transfer (e.g., an abrupt shift in the local sequence identity to the different other genomes in the alignment). Clusters of phage isolates observed in these phylogenies generally correlated very well with established taxonomy as inferred from NCBI taxonomy (https://www.ncbi.nlm.nih.gov/taxonomy), the ICTV (https://talk.ictvonline.org/taxonomy/; [144]), and other reports in the literature [28]. To classify our phage isolates on the species level, we generated pairwise whole-genome alignments generated using MAFFT v7.450 [136] implemented in Geneious 2021.0.1 with genomes of related phages as identified in our phylogenies. From these alignments, the nucleotide sequence identity was determined as the query coverage multiplied by the identity of aligned segments [22]. When the genomes were largely syntenic and showed >95% nucleotide sequence identity, we classified our isolates as the same species as their close relative [143], at least if this phage has been assigned to a species by the ICTV [144].

Most phages were named in honor of scientists and other historically relevant personalities with links to the city of Basel, Switzerland, where the majority of phages had been isolated. Despite efforts to name many phages after female personalities, a considerable gender imbalance remains as a consequence of biases in how science and history have been made and recorded before the 20th century.

## Sequence alignments and phylogenetic analyses

The NCBI GenBank accession numbers of all previously published genomes used in this study are listed in S6 Table. Sequence alignments of different sets of homologous genes were generated using MAFFT v7.450 implemented in Geneious Prime 2021.0.1 [136]. Whenever

required, poor or missing annotations in bacteriophage genomes downloaded from NCBI GenBank were supplemented using the ORF finder tool of Geneious Prime 2021.0.1 guided by orthologous sequence parts of related genomes. Alignments were set up using default settings typically with the fast FFT-NS-2 algorithm and 200 PAM/k = 2 or BLOSUM62 scoring matrices for nucleotide and amino acid sequences, respectively. Subsequently, alignments were curated manually to improve poorly aligned sequence stretches and to mask nonhomologous parts.

For phylogenetic analyses, sequence alignments of orthologous stretches from different genomes (genes, proteins, or whole genome) were used to calculate Maximum-Likelihood phylogenies with PhyML 3.3.20180621 implemented in Geneious Prime 2021.0.1 [145]. Phylogenies were calculated with the HYK85 substitution model for nucleotide sequences and with the LG substitution model for amino acid sequences. For the inference of phylogenetic relationships between phage genomes, we sometimes used curated whole-genome alignments, but the infestation with homing endonucleases and the associated gene conversion made this approach impossible for all larger genomes. Instead, we typically used curated sequence alignments of several conserved core genes (on nucleotide or amino acid level, depending on the distance between the genomes) as the basis for Maximum-Likelihood phylogenies. The detailed procedures for each phylogeny shown in this article are described in S3 Text.

## LPS structures of enterobacterial strains

*Escherichia* K-12 MG1655 (serogroup A) codes for a K12-type LPS core and an O16-type O-antigen, but functional expression of the O-antigen was lost during laboratory adaptation due to inactivation of *wbbL* by an IS5 insertion so that only a single terminal GlcNAc is attached to the LPS core [42,71] (see Fig 2A). *E. coli* strains UTI89 (serogroup B2), CFT073 (serogroup B2), and 55989 (serogroup B1) express O-antigens of the O18, O6, and O104 types, respectively [123,124]. Their core LPS types were determined to be R1 (UTI89 and CFT073) and R3 (55989) by BLAST searches with diagnostic marker genes similar to the PCR-based approached described previously [71]. For the illustration in Fig 11, the structures of LPS cores and linked O-antigen polysaccharides were drawn as described in the literature with typical O-antigen chain lengths of 15 to 25 repeat units [71,146,147]. For the O18-type of O-antigen polysaccharides, different subtypes with slight differences in the repeat unit have been described [147], but, to the best of our knowledge, it has remained elusive which subtype is expressed by *E. coli* UTI89. In Fig 11, we therefore chose an O18A type as exemplary O-antigen polysaccharide for *E. coli* UTI89. Similar to the laboratory adaptation of *E. coli* K-12 MG1655, strains of the *E. coli* B lineage like *E. coli* B REL606 lack O-antigen expression due to an IS1 insertion in *wbbD* but have an additional truncation in their R1-type LPS core due to a second IS1 insertion in *waaT* that leaves only 2 glucoses in the outer core (see Fig 11) [66,98]. *S. enterica* subsp. *enterica* serovar Typhimurium strains 12023s (also known as ATCC 14028) and SL1344 share the common *S.* Typhimurium LPS core and *Salmonella* serogroup B/O4 O-antigen [40,148] (see Fig 11).

## Bacterial genome sequencing and analyses to identify host receptors

While top agar assays with *E. coli* mutants carrying defined deletions in genes coding for different surface proteins or genes involved in the LPS core biosynthesis readily identified the host receptor of most bacteriophages (see above), few small siphoviruses of the Drexlerviridae family and the *Dhillonvirus*, *Nonagvirus*, and *Seuratvirus* genera of the Siphoviridae family could not be assigned to any surface protein as secondary receptor. This was the case for phages AugustePiccard (Bas01) and JeanPiccard (Bas02) of Drexlerviridae, TheodorHerzl

(Bas14) and Oekolampad (Bas18) of *Dhillonvirus*, ChristophMerian (Bas19) and FritzHoffmann (Bas20) of *Nonagvirus*, and VogelGryff (Bas25) of *Seuratvirus*. However, as siphoviruses infecting gram-negative bacteria, it seemed highly likely that these phages would use an outer membrane porin as their secondary receptor [38]. We therefore isolated resistant mutants of *E. coli* K-12 BW25113 by plating bacteria on LB agar plates, which had been densely covered with high-titer lysates of each of these bacteriophages. While attempting to verify that the resistance of isolated bacterial clones was specific to the phage on which they had been isolated, we found that many clones were fully resistant specifically but without exception to all these small siphoviruses with yet no known host receptor, suggesting that they all targeted the same receptor.

For several of these phage-resistant clones, genomic DNA was prepared using the GenElute Bacterial Genomic DNA Kit (Sigma-Aldrich, St. Louis, Missouri, USA) according to the manufacturer's guidelines and sequenced at the Microbial Genome Sequencing Center (MiGS) using the Illumina NextSeq 550 platform. Sequencing reads were assembled to the *E. coli* K-12 MG1655 reference genome (NCBI GenBank accession U000096.3) using Geneious Prime 2021.0.1 with a coverage of typically 100 to 200× to identify the mutations underlying their phage resistance. These analyses uncovered a diversity of in-frame deletions in *lptD* for the different mutant clones (Fig 4).

## Morphological analyses by transmission electron microscopy

The virion morphology of representative bacteriophages of the different taxonomic groups included in the BASEL collection was analyzed by transmission electron microscopy following common procedures in the field [149]. Briefly, 5 μl drops of high-titer lysates were adsorbed to 400 mesh carbon-coated grids, which were rendered hydrophilic using a glow-discharger at low vacuum conditions. They were subsequently stained on 5 μl drops of 2% (w/v) uranyl acetate. Samples were examined using an FEI Tecnai G2 Spirit transmission electron microscope (FEI Company, Hillsboro, Oregon, USA) operating at 80-kV accelerating voltage. Images were recorded with a side-mounted Olympus Veleta CCD camera 4k using EMSIS RADIUS software at a nominal magnification of typically 150,000×.

## Quantification and statistical analysis

Quantitative data sets were analyzed by calculating mean and standard deviation of at least 3 independent biological replicates for each experiment. Detailed information about replicates and statistical analyses for each experiment is provided in the figure legends.

## Supporting information

**S1 Table. List of all bacterial strains used in this study.** The abbreviations in the selection column indicate the drug and its concentration that were used. Amp, ampicillin; Cam, chloramphenicol; Kan, kanamycin; Zeo, zeocin; 25/50/100 refer to 25 μg/ml, 50 μg/ml, and 100 μg/ml, respectively. The following mutants of the KEIO collection were used for qualitative top agar assays but are not included in the strain list because no phage showed any growth phenotype on them: *ompW*::kanR, *phoE*::kanR, *flgG*::kanR, *fepA*::kanR, *hofQ*::kanR, *cirA*::kanR, *fhuE*::kanR, *fiu*::kanR, *ompN*::kanR, *pgaA*::kanR, *chiP*::kanR, *ompL*::kanR, *yddB*::kanR, *fecA*::kanR, *uidC*::kanR, *nanC*::kanR, *yfaZ*::kanR, *bglH*::kanR, *bcsC*::kanR, *cusC*::kanR, *gfcE*::kanR, *mdtP*::kanR, *ompG*::kanR, *ompX*::kanR, *yfeN*::kanR, *csgF*::kanR, *wza*::kanR, *flu*::kanR, *nmpC*::kanR, *eaeH*::kanR, *ydiY*::kanR, *yiaT*::kanR, *yaiO*::kanR, *mdtQ*::kanR, *pgaB*::kanR, *mipA*::kanR, *pldA*::kanR, *yzcX*::kanR, *ydeT*::kanR, *blc*::kanR, *gspD*::kanR, *yjgL*::kanR.
(DOCX)

**S2 Table. List of all oligonucleotide primers used in this study.**
(DOCX)

**S3 Table. List of all plasmids used in this study.** The abbreviations in the selection column indicate the drug and its concentration that were used. Amp, ampicillin; Cam, chloramphenicol; Kan, kanamycin; Zeo, zeocin; 25/50/100 refer to 25 μg/ml, 50 μg/ml, and 100 μg/ml, respectively.
(DOCX)

**S4 Table. Construction of all plasmids generated in this study.**
(XLSX)

**S5 Table. List of all phages used in this study.** The table lists all details regarding the taxonomic classification, isolation/source, host receptors, and genomic features of all phages used in this study. As Caudovirales, all these phages are classified as Viruses/Duplodnaviria/Heunggongvirae/Uroviricota/Caudoviricetes/Caudovirales, so only the classification on the level of family and below as defined by the current viral taxonomy of the ICTV (https://talk.ictvonline.org/taxonomy/; [58,144]) is listed. For the reference phages, genome sizes and estimation of RM system recognition sites are based on the reference genomes in NCBI GenBank as indicated. Note that for phage T5, the reference genome includes the 10,219 bp terminal repeat twice, unlike all other Markadamsvirinae genomes that we listed. The P2*vir* phage had been isolated as a spontaneous mutant of a P2 *cox3* prophage (supposedly unable to excise [151]) and was analyzed by whole-genome sequencing (as described in Materials and methods), revealing few single-nucleotide differences to the reference genome, the expected *cox3* mutation, and a 1-bp insertion in the lysogeny repressor gene *C* that results in a premature stop codon. Recognition sites of RM systems were identified using Geneious Prime 2021.0.1. The numbers include the 2 orientations of palindromic recognition sequences separately and do not account for the fact that the counts are not fully comparable, e.g., because type III RM systems require 2 unmodified recognition sites in head-to-head orientation for cleavage [47]. Given the high number of type III RM recognition sites in all genomes because of their short length (5 nt in case of both EcoCFT_II and EcoP1_I, see Fig 2B), this limitation is largely theoretical. ICTV, International Committee on the Taxonomy of Viruses; RM, restriction–modification.
(XLSX)

**S6 Table. List of all phage genomes used in the in silico analyses.**
(XLSX)

**S1 Data. Raw data and calculations of all EOP assays and qualitative top agar experiments.** EOP, efficiency of plating.
(XLSX)

**S1 Text. Construction of bacterial mutant strains.**
(DOCX)

**S2 Text. Different considerations regarding the composition of the BASEL collection and the phenotypic analyses to characterize it.** BASEL, BActeriophage SElection for your Laboratory.
(DOCX)

**S3 Text. Generation of the Maximum-Likelihood phylogenies shown in this article.**
(DOCX)

**S1 Fig. Supporting information for Figs 3 and 4. (A)** Maximum-Likelihood phylogeny of Drexlerviridae and the *Dhillonvirus* genus of Siphoviridae based on several core genes with bootstrap support of branches shown if >70/100. It is clearly apparent that Drexlerviridae are split into 2 major clades, one formed by Braunvirinae and Rogunavirinae and another one formed by Tempevirinae, Tunavirinae, plus a few other groups. Given that the phylogenies strongly agree on all major branches, the root of the Drexlerviridae phylogeny shown in Fig 3D was placed between these 2 major clades. **(B)** The bona fide RBP loci downstream of the *gpJ* homolog are shown for all small siphoviruses (Drexlerviridae and Siphoviridae of *Dhillonvirus*, *Nonagvirus*, and *Seuratvirus* genera) where we had experimentally determined the terminal receptor (together with selected representatives where previous work had determined the receptor specificity). **(C)** The locus encoding lateral tail fibers was analyzed in a sequence alignment of the 13 Drexlerviridae phage genomes of the BASEL collection (see Materials and methods). It is clearly visible that the upstream and downstream regions (encoding genes involved in recombination as well as primase/helicase proteins for genome replication) are highly conserved and fully syntenic, with exception of small insertions in a few sequences. Conversely, only the most 5′ end of the largest lateral tail fiber protein gene is very similar among all analyzed genomes (green circle), while the rest shows neither synteny nor clear homology across all genomes. **(D)** The bona fide RBP locus of *E. coli* phage RTP was aligned to the homologous locus of phage AugustePiccard (Bas01) as described in Materials and methods. For the region comprising *rtp44* and *rtp45* of phage RTP, the pairwise identity of the 2 nucleotide sequences is ca. 93%. BASEL, BActeriophage SElection for your Laboratory; RBP, receptor-binding protein.
(TIF)

**S2 Fig. Supporting information for Fig 5. (A)** The locus encoding lateral tail fibers was analyzed in sequence alignments of the 5 *Dhillonvirus* phage genomes of the BASEL collection (top) and the 7 Queuovirinae phage genomes (genera *Nonagvirus* + *Seuratvirus*; bottom) of the BASEL collection as described in Materials and methods. In both cases, 2 clear dips in overall sequence similarity are obvious, once at the bona fide RBP locus and then at the lateral tail fiber locus downstream of the far 5′ end of its first gene. **(B)** Schematic comparison of representative bona fide RBP loci as shown in S1B Fig to the corresponding allele of *E. coli* phages JenK1, HdK1, and HdsG1 that does not clearly match any of them. **(C)** *Nonagvirus* and *Seuratvirus* phages share a core 7-deazaguanosine biosynthesis pathway involving FolE, QueD, QueE, and QueC, which synthesizes dPreQ$_0$ that is inserted into their genomes by DpdA. In *Nonagvirus* phages, the fusion of QueC with a Gat domain to Gat-QueC results in the modification with dG$^+$ instead of dPreQ$_0$ [43]. BASEL, BActeriophage SElection for your Laboratory; Gat, glutamate amidotransferase; RBP, receptor-binding protein.
(TIF)

**S3 Fig. Supporting information for Fig 6. (A)** The locus encoding lateral tail fibers was analyzed in a sequence alignment of the Demerecviridae: Markadamsvirinae phage genomes of the BASEL collection as described in Materials and methods. Sequence identity is high upstream and downstream of the lateral tail fiber locus (with exception of presence/absence of a few putative homing endonucleases) but drops considerably at the lateral tail fiber genes. Note that, as described previously, the lateral tail fibers can either be composed of a single large polypeptide or by 2 (or more) separate proteins [49,59]. The same diversity in architecture of lateral tail fibers can also be seen at the corresponding loci of small siphoviruses (S1C and S2A Figs). **(B)** The illustration shows a phylogeny of the RBPs of all Markadamsvirinae phages shown in Fig 6C. Briefly, the RBP genes of all genomes (invariably encoded directly

upstream of the terminase genes) were translated, aligned, and then used to generate a phylogeny as described in Materials and methods. Three clear clusters emerge, one including all phages known to bind BtuB (left), one including all phages known to bind FepA (top right), and one including all phages known to bind FhuA (bottom right). Based on similar analyses by others [61], we conclude that the position of RBPs in these clusters is predictive of terminal receptor specificity of the phages encoding them. BASEL, BActeriophage SElection for your Laboratory; RBP, receptor-binding protein.
(TIF)

**S4 Fig. Supporting information for [Fig 7](A) Top agar assays with different surface protein mutants of the KEIO collection in comparison to the ancestral *E. coli* K-12 BW25113 strain were performed with serial 10-fold dilutions of all Tevenvirinae phages used in this study (undiluted phage lysates at the bottom and increasingly diluted samples toward the top). The phages show impaired growth on each one of the mutants, which identifies the primary receptor of each phage (also indicated by the color code highlighted on the right). Note that growth inhibition on the primary receptor mutants is rarely total and often still enables strongly reduced, heterogeneous plaque formation (arrows), especially after prolonged incubation of the top agar plates. **(B)** The Maximum-Likelihood phylogeny of Tevenvirinae short tail fiber proteins reveals 2 homologous, yet clearly distinct, clusters that correlate with the absence (variant #1, like T4) or presence (variant #2) of detectable LPS core dependence as shown in [Fig 7E](C) The results of (B) indicate that variant #1, as shown for T4, binds the deep lipid A–Kdo region of the enterobacterial LPS core, while variant #2 binds a more distal part of the (probably inner) core. LPS, lipopolysaccharide.
(TIF)

**S5 Fig. Supporting information for Figs [7](–[10](A)** Top agar assays of reference strain *E. coli* K-12 ΔRM carrying empty vector pBR322_ΔP*tet* or pAH213_rexAB were performed with serial 10-fold dilutions of phage T4 wild type, a T4 variant encoding an apparently hypomorphic *rIIAB* fusion (see (B)), and phage T5 (as control). The *rIIAB* mutant of phage T4 is unable to form plaques on the *rexAB*-expressing host and shows only "lysis from without" [97], while the T4 wild type and phage T5 are not affected. **(B)** A T4 phage mutant was erroneously obtained from a culture collection instead of the wild type and encoded a peculiar *rII* allele with fusion of the *rIIA* and *rIIB* open reading frames (shown as orange arrows; genome sequence determined by whole-genome sequencing as described in Materials and methods). Since such a mutant seems unlikely to arise spontaneously during shipping, we find it likely that this phage strain is related to the *rIIAB* fusion mutants employed for discovery of the triplet nature of the genetic code that were once commonly used (concisely reviewed in [152,153]). Notably, position and size of the deletion fusing *rIIA* and *rIIB* are indistinguishable from the sketch drawn by Benzer and Champe for the *rIIAB* fusion mutant *r1589*, which was used in the aforementioned work [154]. Unlike T4 wild type, the *rIIAB* fusion mutant is susceptible to *rexAB* when ectopically expressed from a plasmid vector (see (A)) and therefore validates functionality of the *rexAB* construct. **(C)** The illustration shows a sequence alignment of the locus encoding lateral tail fiber genes in phage rV5 and new isolates DrSchubert, Alex-Boehm, and JeffSchatz that broadly cover the phylogenetic range of this genus ([Fig 8C](It extends from the tail tape measure protein of phage rV5 (*gp49*) to the last large lateral tail fiber gene (*gp27*) [69,70]. Note that most genes are highly conserved including the lateral tail fiber component with sugar deacetylase domain (see (F); around position 8,000 in this alignment) or the short tail fiber locus (compare (D)). Only the 3 largest lateral tail fiber genes show considerable allelic variation (highlighted by arrows) that is very strong for 2 of them (positions 10,000 to 14,000 and 16,000 to 19,000) and moderate for another one (positions 29,000 to

34,000). **(D)** Vequintavirinae sensu stricto (represented by rV5 and Jeff Schatz) encode 2 para-logous short tail fiber proteins and a tail fiber chaperone that are homologous to the corresponding locus in phi92-like phages like PaulScherrer and, ultimately, to short tail fiber GpS and chaperone GpU of Mu(+) (which targets a different glucose in the K12-type LPS core GpS [94,95]). **(E)** Amino acid sequence alignment and Maximum-Likelihood phylogeny of Gp1.2 orthologs in all tested Autographiviridae phages. Phage JeanTinguely (Bas64) belongs to the *Teseptimavirus* genus but encodes an allele of *gp1.2* that is closely related to those of the *Berlinvirus* genus, possibly explaining its resistance to PifA (see Fig 9F). **(F)** Amino acid sequence alignment of the lateral tail fiber Gp64 of phage N4 and a lateral tail fiber conserved among Vequintavirinae and relatives (representatives shown). The proteins share a predicted poly-GlcNAc deacetylase domain as identified by Phyre2 [138]. LPS, lipopolysaccharide.
(TIF)

**S6 Fig. Quantification of bacteriophages that depend on the O16-type O-antigen. (A)** Scheme showing how the fraction of bacteriophages depending on the O16-type O-antigen for each sewage treatment facility was quantified. Briefly, per facility, we first sampled 144 new phage isolates on *E. coli* K-12 MG1655 ΔRM *wbbL(+)* that expresses the O16-type O-antigen (Fig 2A). Subsequently, we performed qualitative top agar spot assays of these isolates on the same strain (growth control) and the parental *E. coli* K-12 MG1655 ΔRM that lacks O16-type O-antigen expression (Fig 2A). O-antigen–dependent isolates were be identified as those phages able to grow on the *wbbL(+)* host but not at all on the parental strain without O-antigen. **(B)** The fraction of bacteriophages that require the O16-type O-antigen for infectivity was very low for all sewage treatment facilities. Precise counts were 7.6% for ARA Basel (11/144), 6.3% for ARA Canius (9/144), and 2.1% for each REAL Luzern and ARA Triengen (3/144 in both cases).
(TIF)

**S7 Fig. Confirming the identity of KEIO collection strains and isogenic mutants. (A, B)** The identity of all KEIO collection strains and isogenic mutants (*btuB* and *tolC* KOs; see Materials and methods and S1 Text) was probed by diagnostic PCR over the gene that should have been replaced with a kanamycin resistance cassette [118] and subsequent agarose gel electrophoresis of the PCR products. Panel (A) shows the agarose gel with all PCR products and, for reference, the "1 kb DNA ladder" of New England Biolabs (right). Panel (B) lists the oligonucleotide primers used for all PCRs (sequences compiled in S2 Table) and the expected sizes of PCR products for the OS PCRs of WT and KO alleles. In cases where WT and KO alleles were expected to have a similar size, we performed an additional PCR with 1 INT primer that anneals inside the kanamycin resistance cassette and 1 primer at the target locus. These PCRs can only result in a PCR product if the correct mutant strain was used as PCR template. The agarose gel in (A) confirms the identity of all KEIO collection KOs and isogenic strains that we used in our study. INT, internal; KO, knockout; OS, overspanning; WT, wild-type.
(TIF)

**S8 Fig. Complementation of all *E. coli* mutants used for bacteriophage receptor identification—*E. coli* K-12 BW25113 wild-type control.** Thirteen phages were chosen to confirm the phenotypes of all *E. coli* mutants used for bacteriophage receptor identification by genetic complementation in qualitative top agar spot assays (S9–S13 Figs). These were N4 (bona fide ECA-targeting phage dependent on *wecB*), TrudiRoth (Bas30; targets BtuB), DanielBernoulli (Bas08; targets TolC), T5 (targets FhuA), IsaakIselin (Bas10; targets YncD), KurtStettler (Bas22; targets LamB), WilhelmHis (Bas35; targets Tsx), T2 (targets FadL), PaulHMueller (Bas45; targets OmpA), FelixPlatter (Bas40; targets OmpC), AlbertHofmann (Bas47; targets

OmpF), VogelGryff (Bas25; targets LptD), and EmilieFrey (Bas61; bona fide ECA-targeting phage dependent on *wecB*). Several phages additionally depend on the LPS core and therefore on *waaC* and sometimes also *waaG* for their infectivity (see Fig 2A). These were DanielBernoulli (Bas08), WilhelmHis (Bas35), AlbertHofmann (Bas47), VogelGryff (Bas25), and EmilieFrey (Bas61). The top agar plate shown in this figure was generated with the parental *E. coli* K-12 BW25113 wild-type control that was readily infected by all phages. LPS, lipopolysaccharide. (TIF)

**S9 Fig. Complementation of *btuB*, *tolC*, and *fhuA* mutants.** Top agar spot assays with the 13 phages highlighted in S8 Fig were performed with the *btuB* (top), *tolC* (middle), and *fhuA* (bottom) mutant strains of *E. coli* K-12 BW25113. The plates on the left side were generated with strains carrying the respective complementation plasmid (see Materials and methods) and the plates on the right side with *E. coli* carrying the ev control. The plates show that the *btuB* dependency of TrudiRoth (Bas30; top), the *tolC* dependency of DanielBernoulli (Bas08; middle), and the *fhuA* dependency of T5 (bottom) could be fully complemented. ev, empty vector. (TIF)

**S10 Fig. Complementation of *yncD*, *lamB*, and *tsx* mutants.** Top agar spot assays with the 13 phages highlighted in S8 Fig were performed with the *yncD* (top), *lamB* (middle), and *tsx* (bottom) mutant strains of *E. coli* K-12 BW25113. The plates on the left side were generated with strains carrying the respective complementation plasmid (see Materials and methods) and the plates on the right side with *E. coli* carrying the ev control. The plates show that the *yncD* dependency of IsaakIselin (Bas10; top), the *lamB* dependency of KurtStettler (Bas33; middle), and the *tsx* dependency of WilhelmHis (Bas35; bottom) could be fully complemented. ev, empty vector. (TIF)

**S11 Fig. Complementation of *ompA*, *ompC*, and *waaC* mutants.** Top agar spot assays with the 13 phages highlighted in S8 Fig were performed with the *ompA* (top) and *ompC* (middle) mutant strains of *E. coli* K-12 BW25113 as well as the *waaC* mutant of *E. coli* K 12 MG1655 ΔRM (bottom). The plates on the left side were generated with strains carrying the respective complementation plasmid (see Materials and methods) and the plates on the right side with *E. coli* carrying the ev control. The plates show that the *ompA* dependency of PaulHMueller (Bas45; top) and the *ompC* dependency of FelixPlatter (Bas40; middle) could be fully complemented. Similarly, the loss of infectivity of DanielBernoulli (Bas08), WilhelmHis (Bas35), AlbertHofmann (Bas47), VogelGryff (Bas25), and EmilieFrey (Bas61) on the *waaC* mutant could be fully complemented by genetic complementation (bottom). ev, empty vector. (TIF)

**S12 Fig. Complementation of *waaG*, *wecB*, and *fadL* mutants.** Top agar spot assays with the 13 phages highlighted in S8 Fig were performed with the *waaG* mutant of *E. coli* K 12 MG1655 ΔRM (top) and the *wecB* (middle) as well as *fadL* (bottom) mutants of *E. coli* K-12 BW25113. The plates on the left side were generated with strains carrying the respective complementation plasmid (see Materials and methods) and the plates on the right side with *E. coli* carrying the ev control. The plates show that the total loss of infectivity of EmilieFrey (Bas61) as well as the clearly reduced infectivity of AlbertHofmann (Bas47) and VogelGryff (Bas25) on the *waaG* mutant could be fully complemented by genetic complementation (top). Similarly, the *wecB* dependency of N4 (middle) and the *fadL* dependency of T2 (bottom) could be fully complemented. Unlike for all other complementation constructs, full complementation with the *wecB* plasmid required induction of expression driven by the P*lac* on pAH186SC101 with 1 mM of IPTG during the overnight culture. ev, empty vector; IPTG, isopropyl β-d-

1-thiogalactopyranoside.
(TIF)

**S13 Fig. Complementation of *ompF* and *lptD* mutants.** Top agar spot assays with the 13 phages highlighted in S8 Fig were performed with the *ompF* knockout (top) as well as the *lptDΔ(Y658-Y678)::H* (middle) and *lptDΔ(L394-V396)::Y* mutants (middle and bottom plates; see also Fig 4C and 4D). The plates on the left side were generated with strains carrying the respective complementation plasmid (see Materials and methods) and the plates on the right side with *E. coli* carrying the ev control. The plates show that the *ompF* dependency of AlbertHofmann (Bas47; top) and the *lptD* dependency of VogelGryff (Bas25; middle and bottom plates) could be fully complemented. Curiously, complementation of the *ompF* mutant abolished infections by FelixPlatter (Bas40; compare left and right plates of the top panel). A similar phenomenon was previously observed by others who reported that *ompF* overexpression inhibited infections by bacteriophage T4, possibly by indirectly interfering with expression of its primary receptor OmpC that is also the primary receptor of FelixPlatter (Bas40; see Figs 7C and 7D and S4A) ([72] and literature cited therein). ev, empty vector.
(TIF)

## Acknowledgments

The authors are grateful to Dr. Julie Sollier, Prof. Martin Loessner, Prof. Urs Jenal, and Prof. Marek Basler for valuable input and critical reading of the manuscript. We are deeply indebted to Prof. Urs Jenal and Prof. Christoph Dehio for sharing multiple different *E. coli* strains and plasmids (S1 and S3 Tables). Prof. Calin Guet is acknowledged for sharing plasmids (S3 Table). The authors thank high school students participating in the Biozentrum Summer Science Academy 2019, Julia Harms, and Dr. Thomas Schubert for assistance with the isolation of multiple bacteriophages described in this study (S5 Table). *Salmonella* Typhimurium strains 12023s and SL1344 were obtained from Prof. Dirk Bumann and Prof. Mederic Diard, respectively, while *E. coli* B REL606 was obtained from Dr. Jenna Gallie. Phage T4D and the P1 lysogen *E. coli* EB1484 were obtained from Prof. Kenneth Kreuzer. Phage P2*vir* was generously shared by Dr. Nicolas Wenner and Prof. Jay Hinton. The Coli Genetic Stock Center (CGSC) and the German Collection of Microorganisms and Cell Cultures (DSMZ) are acknowledged for sharing different strains of *E. coli* or different bacteriophages, respectively (S1 and S5 Tables). We thank Dr. Mohamed Chami and Carola Alampi of the BioEM lab of the Biozentrum, University of Basel, for their support with transmission electron microscopy. The authors are grateful to Dr. Nikoline Olsen and Prof. Bert van den Berg for helpful comments regarding the genetic basis of phage resistance to host immunity and regarding the use of porin channels as bacteriophage receptors, respectively. Dr. Christian Beisel and Dr. Geoffrey Fucile are acknowledged for advice regarding DNA sequencing. The authors are grateful to ARA Basel, ARA Canius (Lenzerheide), ARA Limeco (Dietikon), ARA Triengen, and REAL Luzern (Emmen) for providing samples of sewage plant inflow and to Universitätsspital Basel for hospital sewage.

## Author Contributions

**Conceptualization:** Alexander Harms.

**Data curation:** Alexander Harms.

**Formal analysis:** Enea Maffei, Aisylu Shaidullina, Marco Burkolter, Yannik Heyer, Fabienne Estermann, Valentin Druelle, Patrick Sauer, Luc Willi, Sarah Michaelis, Hubert Hilbi, David S. Thaler, Alexander Harms.

**Funding acquisition:** Aisylu Shaidullina, Valentin Druelle, Hubert Hilbi, Alexander Harms.

**Investigation:** Enea Maffei, Aisylu Shaidullina, Marco Burkolter, Yannik Heyer, Fabienne Estermann, Valentin Druelle, Patrick Sauer, Luc Willi, Sarah Michaelis, Hubert Hilbi, David S. Thaler, Alexander Harms.

**Methodology:** Enea Maffei, Aisylu Shaidullina, Marco Burkolter, Yannik Heyer, Fabienne Estermann, Valentin Druelle, Patrick Sauer, Luc Willi, Sarah Michaelis, Hubert Hilbi, David S. Thaler, Alexander Harms.

**Project administration:** Alexander Harms.

**Resources:** Enea Maffei, Aisylu Shaidullina, Marco Burkolter, Yannik Heyer, Fabienne Estermann, Valentin Druelle, Patrick Sauer, Luc Willi, Sarah Michaelis, Hubert Hilbi, David S. Thaler, Alexander Harms.

**Software:** Valentin Druelle, Alexander Harms.

**Supervision:** Enea Maffei, Fabienne Estermann, Hubert Hilbi, David S. Thaler, Alexander Harms.

**Validation:** Enea Maffei, Aisylu Shaidullina, Marco Burkolter, Yannik Heyer, Fabienne Estermann, Valentin Druelle, Patrick Sauer, Luc Willi, Sarah Michaelis, Hubert Hilbi, David S. Thaler, Alexander Harms.

**Visualization:** Fabienne Estermann, Alexander Harms.

**Writing – original draft:** Alexander Harms.

**Writing – review & editing:** Enea Maffei, Aisylu Shaidullina, Marco Burkolter, Yannik Heyer, Fabienne Estermann, Valentin Druelle, Patrick Sauer, Luc Willi, Sarah Michaelis, Hubert Hilbi, David S. Thaler, Alexander Harms.

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
