## [Editor Report · Decision Letter 0]

20 Mar 2021

Dear Dr. Harms, 

Thank you for submitting your manuscript entitled "Systematic exploration of Escherichia coli phage-host interactions with the BASEL phage collection" for consideration as a Methods and Resources by PLOS Biology.

Your manuscript has now been evaluated by the PLOS Biology editorial staff, as well as by an academic editor with relevant expertise, and I am writing to let you know that we would like to send your submission out for external peer review.

Please re-submit your manuscript within two working days, i.e. by Mar 22 2021 11:59PM.

Kind regards,

Paula

---

Associate Editor

PLOS Biology

---

## [Decision Letter · Decision Letter 1]

6 May 2021

Dear Dr. Harms,

Thank you very much for submitting your manuscript "Systematic exploration of Escherichia coli phage-host interactions with the BASEL phage collection" for consideration as a Methods and Resources at PLOS Biology. Your manuscript has been evaluated by the PLOS Biology editors, an Academic Editor with relevant expertise, and by several independent reviewers.

In light of the reviews (below), we will not be able to accept the current version of the manuscript, but we would welcome re-submission of a much-revised version that takes into account the reviewers' comments. We cannot make any decision about publication until we have seen the revised manuscript and your response to the reviewers' comments. Your revised manuscript is also likely to be sent for further evaluation by the reviewers.

In particular, reviewer #1 says that the figures and text are far too dense to appreciate what has been done, and that the paper is overwhelming and difficult to read. Reviewer #1 and #3 have concerns for the use of a mutant with no O-antigen. Reviewer #2 asks whether the Keio collection mutants were confirmed for their genotypes, and points that the results in TolC mutants should be interpreted with caution. Reviewer #2 and #3 ask or the complementation of single gene mutants. Reviewer #3 says that the isolation of the phages on K-12 is a possible limitation to the actual mechanistic diversity of the phage found, that the discussion of trade-offs is confusing and it is not clear your measurements demonstrate any real trade-offs, says that you do not provide support to some of your claims, says that you should add TEM images of the phages, and asks about the generalization of the work. This reviewer considers that addressing these questions and the host range of phages, adding also a systematic discussion on the utility of such broadened phage collection would be highly valuable. Please address all the reviewers' concerns. 

We expect to receive your revised manuscript within 3 months. 

**IMPORTANT - SUBMITTING YOUR REVISION**

*Re-submission Checklist*

*Published Peer Review*

*PLOS Data Policy*

*Blot and Gel Data Policy*

Sincerely,

Paula

---

Paula Jauregui, PhD

Associate Editor

PLOS Biology

REVIEWS:

Reviewer #1: Phage therapy.

Reviewer #2: Biology and application of phages.

Reviewer #3: Systems biology.

Reviewer #1: As reviewers, we are supportive of the intent of the paper; however, we do not feel that there is a conceptual advance made that warrants publication in PLOS Biology. The goal of developing a diverse resource collection that can be interrogated to inform basic phage biology is commendable. Unfortunately, in its current presentation - the figures and text are far too dense to appreciate what has been done. There probably are important insights in this paper, but it is too difficult to find these insights. The paper is overwhelming and difficult to read. One major barrier to the readability is the nomenclature - the taxonomy is being used too frequently, given that these words are not intuitive, difficult to say, (and likely will change) I suggest that the authors do not put so much emphasis on the taxonomy.

The first few sections of the results: Composition of BASEL Collection and 2 'overview' sections, do not actually contain results and do not belong here; this information can be distilled down to be included in the intro. 

Figures 1-3 - do not actually contain much data (with the exception of a small panel in fig 2B).

The bait strain here lacked O antigen - this was confusing - this means no phages dependent on O antigen could be isolated. However, they acknowledge that phages may depend on O antigen. Page 6, line 111 - the description of this mutant would suggest you would not get any O-antigen dependent phages, which they did (i.e., the T5-like phages), so I'm just left confused. 

The following comment applies in general to many figures but is using Fig 7 as an example: the data itself in this figure (i.e., Fig 7c) is too small, the phylogenetic tree could be supplemental, I'm not sure about panel A - why is this here? The data pertaining to the receptors (depicted in the far-right panel) is not even shown on this figure - why is a schematic cartoon more important than the actual data? 

Callouts explaining background on figures is odd for a primary research article and were reminiscent of figures more appropriate for a review article. Perhaps the authors can consider publishing a more clear, succinct version of the manuscript focusing on the data collected and provide a commentary piece/review type article elsewhere. 

Reviewer #2: The manuscript by Maffei et al represents a large amount of work on a collection of phages infecting a model organism, E. coli K-12. The Phages are sequenced and classified, and presented as groups using the new ICTV taxonomy. The phages are comprehensively evaluated for their receptor binding protein content, sensitivity to deletions in surface feature genes and anti-phage systems, and host range. A number of common features are noted among phage groups and this work represents a comprehensive and high-impact study that will aid in rationalizing at least a subsection of the large volume of phage genome sequence available. I have several comments and suggestions below that would strengthen the overall work and its presentation.

GENERAL COMMENTS

In the methods, were the Keio collection mutants confirmed for their genotypes (e.g., by PCR)? Keio mutants have a reputation for not always being as advertised.

Also in the methods, were any of the single-gene deletions complemented to confirm the phenotype. This is not described in the methods but is standard practice when studying the phenotypes associated with gene knockouts.

Use caution when interpreting results in mutants with deep LPS core defects. The "deep rough" phenotype has been associated with pleiotropic effects including derangement in LPS structure and proper placement of porins in the outer membrane, which could affect phage sensitivity in multiple ways. This is also true of TolC mutants, which appear to have improperly processed LPS and similar outer membrane defects.

Genbank/INSD accession numbers are not yet available for the phages but these should be made available in the tables provided with the final submission.

SPECIFIC COMMENTS

L 43: use eject instead of inject

L46: use virulent instead of lytic

L127: specify that these are phage isolates

L151: can you provide a citation for phages that use the ECA as a receptor? 

L185: not sure what the use of shortly is here. Breifly?

L227: these bona fide RBP loci

L329: this is imprecise language, the tail contraction may be described as syringe-like but the DNA ejection mechanism is presumably the same as for many sipho and podophages, driven by pressure differential.

L385-386: If ECA was the secondary receptor, would you not expect the change in phage sensitivity to be more drastic? The plating deficiency here seems to mostly be in the range of 1-2 log. Could loss of ECA have other effects on LPS structure that would account for the phenotype? Or is your model that ECA is dispensable?

L 419: ejection

Fig. 1: The podoviral model could also depict a central tail needle structure as observed in phage P22.

Fig. 2: You may wish to label the boxes called Siphoviridae, Myovirdae and Podoviridae as "Other Siphovirdae", "Other Myoviridae" and "Other Podoviridae" as Sipho/Myo/Podo families are likely to be dissolved at some point and the genera they contain will be promoted to other families or sub-families.

All figures: Figure resolution is low, some of the graph axis labels are unreadable. Increase the text font sizes in the graphics where possible and be sure uo provide high-resolution figure in the final version.

Reviewer #3: Systematic exploration of Escherichia coli phage-host interactions with the BASEL phage collection. 

Maffei et al. report on a new valuable, well-characterized collection of coliphage isolated on a particular strain of Escherichia coli K-12. The characterization of the phage by sequence, parameters of infectivity, and interaction with specific phage defense systems is timely and well-done as is placing this information against ten-other well studied phage characterized in parallel. Providing a standard, systematic protocol for phage characterization and providing open access to the phage collection is a great service to the community. Overall, this is a strong paper making an important contribution to the coliphage biology field. We present our concerns and critical feedback below to improve the manuscript. 

Broadly, this team isolates ~100 phages from diverse environmental sources, dissects the critical host molecular mechanisms governing each 66 phages using a collection of smartly chosen E. coli strains and goes so far as to assign possible (sometimes actual) phage molecular determinants as well. The systematic dissection of possible host surface receptors, interactions with E.coli restriction modification (RM) systems, - and abortive infection (Abi) systems for each phage class was elegant and well reasoned for the most part. While some caveats are probably justified due to expression of defense systems on plasmids, we don't think this really would modify the outcomes if done from the genome and indeed might lead to muddier conclusions. Comparison to basic infection on other E. coli and Salmonella strains was welcome.

This said, the isolation of these phage on K-12 strain is a possible limitation to the actual mechanistic diversity of the phage found despite the clear phylogenetic diversity of the collection. K-12 is an rfb mutant and thus has a rough phenotype with a complete LPS core and no O-antigen. It is known, for example, that many phage use the O-antigen as a receptor. That said, they did use a restored O16-antigen bearing K-12 to test their phage, as well as other related variants, at least addressing for their isolated phage what some of their interactions might be (as authors have alluded in S3 Text). 

Their dissection of the different ways the phage could evade the various restriction systems was elegant- largely delineated specifically encoded DNA protection functions and cut-site depletion strategies. Their discussion of trade-offs was however confusing. It wasn't clear their measurements demonstrated any real tradeoffs- just diversity. And the statement that these could be removed by engineering calls into question the idea that these are real tradeoffs in their minds. This we think remains to be tested. 

Authors do not provide support to their claims (line 31-33, 90-93 for example) that this work provides insights into "adaptation of different phage groups to specific niches" or that "These findings greatly expand our understanding of bacteriophage ecology, evolution, and their interplay with bacterial immunity systems". As mentioned earlier, these phages were isolated from diverse sources (though, mainly wastewater streams) on a parent lab strain of E. coli and the pattern of phage sensitivity to different restriction modification and Abi systems expressed in higher copy numbers do not explain the phage ecology, evolutionary pressures in natural environments and natural interplay of phages with all known and unknown bacterial immunity systems. Further, authors seem to claim that the phage infectivity pattern they see here determines the host range of phages (line 721), and we think this is a simplistic take on the complexity of phage host range. We question, is it possible to take these insights on possible phage receptors, sensitivity of phages to RM and Abi systems and predict the phage sensitivity of a non-model natural E coli strain variant. Addressing these questions and a systematic discussion on the utility of such a broadened phage collection would be highly valuable to highlight the importance of the work. 

Finally, authors use a collection of E. coli K-12 single-gene (encoding surface protein) mutants ("KEIO collection") to determine probable receptors of BASEL phage collection and claim that this will serve as a powerful tool for future studies (line 33, 93, 722 for example). We question the generality of such an approach. Can we extend this characterization work to other model strains such as ESKAPE pathogens? What would make such an effort possible? 

Obviously, it is known that there are multiple factors that mediate whether a particular mechanism can affect phage fitness on a host so even relatively close but wild strains of the same bacterium might not show the same susceptibilities or resistances to a model strain. It is a critical point that it has not been entirely proven how to generalize or formulate the results of a screen on a 'tame' collection of mutants like the KEIO collection to other strains much less to different classes of microbe. As raised earlier, the generality of the approach presented in this work itself is also difficult to argue. While it's accepted that the forward genetic methods are an effective method for discovery, they have proven hard to implement in diverse organisms. Even with advancement in newest functional genomic technologies that have come out recently, extending them to different pathogen classes and their natural population diversity such that this data can be effectively used for, (as an example, design of phage therapies) is still, we think, in the distant future. Much of the data presented in this work (and other publications that have recently come out) point to a complex interaction among genetic factors (and environment) that mediate phage/host interactions and while there are certainly conserved themes in the specific modes of these mechanisms and their interactions currently prevent prediction of specific phage/host behaviors. We do believe however, that there may be systematic ways to explore host and phage diversity in nature such that a more generalizable predictive 'model' can be built. But it may require a much larger set of experiments than even this work puts forth. 

We highly recommend addressing these points in the discussion to place the current work in the broader context and why would such an effort be valuable to the field in general. 

Minor Points:

Though we appreciate the effort in making these figures pretty, they are too crowded and make it difficult to read (at least in the pdf we have access to). Here are our suggestions

Fig 1, 2 and 3 can be combined into one, and Fig 1 and 2 need to be simplified. For example, we are not even sure Fig 1 is needed. Fig 2 has phage schematic representative of phages isolated. If they are TEM images that would have been highly valuable. By the way, we thought having a TEM image of phages is one of the gold standard methods of characterization other than sequencing. We find not having TEM images associated with each phage surprising, and posit that those should be part of phage characterization workflows. 

In other figs, schematic of phage infection mode can be minimized by giving more space to the phenotyping data and phylogenetic trees for easy readability. 

Fig 5 can be moved to Suppl info.

Fig 6 panel E can be moved to Suppl info. 

The color coding of each phage name to a possible receptor needs to be mentioned in the legend. 

Fig 12B is complex and non legible, and needs to have higher image space, while LPS schematic can be reduced in size.

As the RM and Abi systems are expressed from plasmids, how does different MOI of phage impact infection patterns? Did authors use different MOI to assess the sensitivity?

Though authors have used single-gene mutants to assess the target receptor, gene complementation studies are not presented. If this work is setting up a standard of phage characterization, then authors need to clarify why not doing those experiments still support their claims similar to the point about TEM image made above.

Authors use "resistance" "sensitivity" "inhibition" "susceptibility" words to describe host and phage interaction. Clarifying their meaning in advance and keeping it consistent throughout would help readers. Further, clarifying the phenotypic data presented in Figs (to LPS mutants, RM and Abi systems) would be helpful. Specifically, clarifying the meaning of lower or higher EOP would be very helpful while going through the detailed text.

---

## [Decision Letter · Decision Letter 2]

15 Sep 2021

Dear Dr. Harms,

Thank you for submitting your revised Methods and Resources entitled "Systematic exploration of Escherichia coli phage-host interactions with the BASEL phage collection" for publication in PLOS Biology. I have now obtained advice from the original reviewers and have discussed their comments with the Academic Editor. 

Based on the reviews, we will probably accept this manuscript for publication, provided you satisfactorily address the following data and other policy-related requests.

DATA POLICY:

Regardless of the method selected, please ensure that you provide the individual numerical values that underlie the summary data displayed in the following figure panels as they are essential for readers to assess your analysis and to reproduce it: Figures 3E, 5F, 6E, 7E, 8D, 9F, 10DG, 12BCDEFGH, S6B.

**Please also ensure that figure legends in your manuscript include information on where the underlying data can be found, and ensure your supplemental data file/s has a legend.**

We expect to receive your revised manuscript within two weeks.

We would also like to ask you whether you have any suggestions for images that we could use as cover for the journal's issue. 

*Published Peer Review History*

*Early Version*

Sincerely,

Paula

---

Associate Editor,

pjaureguionieva@plos.org,

PLOS Biology

Reviewer remarks:

Reviewer #1: I commend the authors on the detailed responses to my comments. My comments have been addressed, I have nothing further

Reviewer #2: Recommends acceptance.

Reviewer #3: The response to reviewers and revision are acceptable. We believe this paper to be an important addition to the phage literature and reagent pool.

---

## [Editor Report · Decision Letter 3]

27 Sep 2021

Dear Dr Harms,

I'm handling your paper temporarily while my colleague Dr Paula Jauregui is out of the office. On behalf of my colleagues and the Academic Editor, Jeremy Barr, I'm pleased to say that we can in principle offer to publish your Methods and Resources "Systematic exploration of Escherichia coli phage-host interactions with the BASEL phage collection" in PLOS Biology, provided you address any remaining formatting and reporting issues. These will be detailed in an email that will follow this letter and that you will usually receive within 2-3 business days, during which time no action is required from you. Please note that we will not be able to formally accept your manuscript and schedule it for publication until you have made the required changes.

PRESS: We frequently collaborate with press offices. If your institution or institutions have a press office, please notify them about your upcoming paper at this point, to enable them to help maximise its impact. If the press office is planning to promote your findings, we would be grateful if they could coordinate with biologypress@plos.org. If you have not yet opted out of the early version process, we ask that you notify us immediately of any press plans so that we may do so on your behalf.

Sincerely, 

Roli Roberts

Roland G Roberts PhD,

Senior Editor

PLOS Biology

rroberts@plos.org

on behalf of

Paula Jauregui, PhD 

Associate Editor 

PLOS Biology
